# Structure and mechanism of the human NHE1-CHP1 complex

Yanli Dong [1,2,4], Yiwei Gao [1,2,4], Alina Ilie[3], DuSik Kim[3], Annie Boucher[3], Bin Li[1,2], Xuejun C. Zhang [1,2✉], John Orlowski [3✉] & Yan Zhao[1,2✉]

Sodium/proton exchanger 1 (NHE1) is an electroneutral secondary active transporter present on the plasma membrane of most mammalian cells and plays critical roles in regulating intracellular pH and volume homeostasis. Calcineurin B-homologous protein 1 (CHP1) is an obligate binding partner that promotes NHE1 biosynthetic maturation, cell surface expression and pH-sensitivity. Dysfunctions of either protein are associated with neurological disorders. Here, we elucidate structures of the human NHE1-CHP1 complex in both inward- and inhibitor (cariporide)-bound outward-facing conformations. We find that NHE1 assembles as a symmetrical homodimer, with each subunit undergoing an elevator-like conformational change during cation exchange. The cryo-EM map reveals the binding site for the NHE1 inhibitor cariporide, illustrating how inhibitors block transport activity. The CHP1 molecule differentially associates with these two conformational states of each NHE1 monomer, and this association difference probably underlies the regulation of NHE1 pH-sensitivity by CHP1.

[1] National Laboratory of Biomacromolecules, CAS Center for Excellence in Biomacromolecules, Institute of Biophysics, Chinese Academy of Sciences, Beijing, China. [2] College of Life Sciences, University of Chinese Academy of Sciences, Beijing, China. [3] Department of Physiology, McGill University, Montreal, QC, Canada. [4] These authors contributed equally: Yanli Dong, Yiwei Gao. ✉email: zhangc@ibp.ac.cn; john.orlowski@mcgill.ca; zhaoy@ibp.ac.cn

Secondary active counter-transport of monovalent cations such as sodium and potassium for protons across biological membranes are prevalent in all kingdoms of life and play vital roles in intracellular pH and cation homeostasis[1,2]. These solute carriers belong to the superfamily of cation/proton antiporters or exchangers and can be broadly segregated into two groups depending on whether they exchange one cation for either one or two protons (electroneutral or electrogenic, respectively). Mammals possess thirteen electroneutral cation/proton antiporters belonging to the solute carrier SLC9 gene family (simply referred to as Na⁺/H⁺ exchangers, NHEs). They display varied tissue distribution, subcellular localization, cation selectivity, and regulation in response to diverse stimuli[3–5]. NHE1 (SLC9A1) is the best characterized member of this family. It is expressed in almost all cell types and localizes to the plasma membrane (basolateral membrane of polarized epithelia) where it preferentially mediates the exchange of extracellular $Na^+$ for intracellular $H^+$ (i.e., $Na^+$-selective NHE)[6,7]. NHE1 plays an essential role in regulating intracellular pH as well as cellular volume, fundamental activities required for the progression of numerous physiological processes underlying cell growth, proliferation, migration, and apoptosis[6]. Dysregulation of NHE1 activity has been implicated in several human disease conditions, including cardiovascular and cerebral injuries associated with ischemia- and reperfusion, cardiac hypertrophy, proliferation and transformation of tumor cells, and neurological disorders[8–14].

Molecular mechanisms of bacterial homologs of the NHEs (termed Nha or Nap) have been studied extensively over the past two decades, including structural studies of electrogenic NhaA from *Escherichia coli* (EcNhaA)[15] and NapA from *Thermus thermophilus* (TtNapA)[16], and electroneutral NhaP from *Pyrococcus abyssi* (PaNhaP)[17], NhaP from *Methanocaldococcus jannaschii* (MjNhaP)[18], and most recently mammalian NHE9 from *Equus caballus*[19]. These studies revealed that these transporters assemble as homodimers, with each monomer possessing 12 or 13 transmembrane (TM) helices that contain a dimerization and a core domain. The ion-translocation pore is located between these two domains with highly conserved negatively charged residues clustered halfway across the lipid bilayer. These acidic residues are involved in cation binding and are alternately exposed to the intracellular and extracellular sides of the membrane during the transport cycle[15–18]. An elevator-like transport model has been proposed for electrogenic Nha/Nap homologs based on structural analyses of both inward- and outward-facing conformations[16,20]. Despite advances in structural studies of prokaryotic NHE homologs, more structural insight is highly desirable to more fully understand the transport mechanism of mammalian NHEs due to their low sequence homology with bacterial homologs. In addition, unlike their bacterial homologs, mammalian NHEs are under strict and multiplexed regulation which adds more complexity to deciphering their mechanisms.

Human NHE1 features a long C-terminal tail that interacts with a variety of cytoplasmic partners involved in modulating its activity at the cell surface[3,6]. For instance, calcineurin B-homologous proteins (CHP1−3) are a family of EF-hand calcium binding proteins that serve as obligatory binding partners and regulators of NHE1 activity[21–24]. In particular, CHP1 is ubiquitously expressed and maintains basal NHE1 activity. Loss-of-function mutations of CHP1 reduced localization of NHE1 at axon terminals and elicited axon degeneration of murine cerebellar Purkinje cells[25] and is associated with human autosomal recessive cerebellar atrophy and ataxia[26,27]. Although an NMR study revealed that CHP1 binds to a juxtamembrane segment (i.e. residues 514−545) of NHE1[11,28], the architecture of the NHE1-CHP1 complex using full-length proteins remains unknown. Since NHE1 is the primary isoform in human cardiac and neuronal cells, a range of inhibitors such as cariporide have been developed and explored for treatment of ischemia and reperfusion injuries[29–31]. However, little structural information is available about how these inhibitors bind and block NHE1 activity.

In this work, we purify the recombinant human NHE1-CHP1 complex, reconstitute it into nanodiscs and determine the complex structures in both an inward-facing conformation and a cariporide-bound outward-facing conformation by using the cryo-electron microscopy (cryo-EM) method. The results provide insights into the regulatory role of CHP1 on NHE1 activity, the inhibitory mechanism by cariporide and details of the alternating-access mechanism for electroneutral sodium/proton exchange.

## Results and discussion

**Structural determination of the NHE1-CHP1 complex.** To elucidate the architecture of the NHE1-CHP1 complex, we co-expressed full-length human NHE1 and CHP1 in HEK293 cells. To exclude a possible distortion bias from a detergent micelle environment, we reconstituted the detergent-solubilized complex into lipid nanodiscs and determined its structure by single-particle cryo-EM (Supplementary Fig. 1a−c). To identify the $Na^+$ binding site, we first prepared the sample in the presence of 150 mM NaCl at pH 7.5, and the corresponding complex structure is termed as NHE1-CHP1$^{Na/7.5}$. In addition to discernable features of transmembrane helix density, two-dimensional (2D) classification also revealed some "crab"-shaped classes featuring splayed and blurred densities protruding from one side of the nanodisc which are presumably due to CHP1 association from the cytoplasm. This initial study also confirmed symmetrical dimerization of NHE1 molecules with the symmetry axis perpendicular to the membrane plane. Initial three-dimensional (3D) classifications, in the presence of C2 symmetry, generated eight classes. Among them, four classes each showed two lumps of weak CHP1 densities located at different positions relative to the central plane of the nanodisc, suggesting conformational mobility of the bound CHP1 molecules relative to the NHE1 dimer or a possibility that the NHE1 dimer was simply not fully occupied by CHP1. One of the four classes featuring clearly resolved transmembrane helices was selected for focused 3D reconstructions of the transmembrane domain (TMD) of NHE1 and yielded an EM map of 3.3-Å resolution, which is rich in features, including densities for sidechains and associated lipid molecules, allowing us to build a de novo model of the NHE1 dimer (Supplementary Fig. 2 and Table 1). However, we were unable to resolve the CHP1 molecules from this small set of particles, probably due to severe conformational heterogeneity.

To investigate how the NHE1-CHP1 complex is activated by protonation, we prepared a sample at pH 6.5 as well as in the presence of 150-mM NaCl. Three-dimensional reconstruction of the corresponding NHE1-CHP1$^{Na/6.5}$ dataset was similar to the NHE1-CHP1$^{Na/7.5}$ dataset, resulting in a 3.4-Å map for the transporter complex (Supplementary Fig. 3 and Table 1). With over 380k complex particles from this dataset, we were able to explore how the CHP1 molecules bind to the NHE1 dimer. We performed symmetry expansion to efficiently analyze every particle regardless of the number of bound CHP1 molecules and carried out CHP1-focused 3D classification on one half of the NHE1 dimer. At this point, 3D classification generated four classes; one of them accounts for 24% of total particles and exhibits a prominent density feature for the bound CHP1. Subsequent 3D reconstruction focusing on this class without an imposed symmetry yielded a 4.0-Å EM map. Mainchain density was clearly resolved for CHP1. Two helices were observed in the cytoplasmic region of the NHE1 protomer, termed HC1 and HC2. Whereas HC1 shows extensive interactions with CHP1,

HC2 is amphipathic and presumably binds to the membrane surface.

To investigate how a pharmacological competitive antagonist blocks NHE1 activity, we included an NHE1 selective inhibitor, cariporide, during protein expression, purification and grid preparation, and also replaced $Na^+$ with non-transportable $K^+$ while maintaining pH at 7.5[32]. During 2D classification, distinct classes were observed showing that cytoplasm-protruding ends of the CHP1 molecules move closer to the membrane plane than those obtained from the NHE1-CHP1[Na/6.5] dataset (Supplementary Fig. 4). Initial 3D classification yielded eight classes, and among them classes 5 and 8 showed continuous TM helical features. In class 5 (~12%), the CHP1 molecules exhibited high mobility and resembled the NHE1-CHP1[Na/7.5] complex, whereas class 8 (~11%) showed two prominent CHP1 molecules lying on the putative intracellular side of the nanodisc and being related by the 2-fold symmetry of the NHE1 dimer. We processed classes 5 and 8 following a similar procedure used for the NHE1-CHP1[Na/7.5] complex, yielding 4.3-Å and 3.5-Å maps, respectively, with all of the TM helices resolved. The resulting 3.5-Å density map clearly revealed hallmark features of both NHE1 and CHP1, and enabled us to unambiguously build a de novo model of the NHE1 dimer and to fit the NMR structure of the subcomplex CHP1-NHE1[514–545] (PDB ID: 2E30)[11] into the map. The CHP1 model agrees well with the EM density, including the secondary structural elements as well as many bulky sidechain residues (Supplementary Fig. 4 and Table 1). One more intracellular helix, HC3, was resolved in each protomer, which is located around the 2-fold axis and in a cleft formed between TMD of the NHE1 dimer and CHP1 molecules. The cariporide molecule was resolved only in the 3.5-Å map, and thus this complex model is termed NHE1-CHP1[K/cariporide].

**Architecture of the NHE1-CHP1 complex.** Human NHE1 consists of 815 amino-acid residues, with an N-terminal transmembrane domain (TMD) and a C-terminal cytoplasmic tail. The intactness of the full-length NHE1 was confirmed by an SDS-PAGE analysis (Supplementary Fig. 1), which shows that the NHE1 sample migrated as two bands of ~95 kDa and ~110 kDa which represent the core- and fully glycosylated forms, respectively, consistent with previous observations[33]. Our NHE1 model is, however, only composed of residues 87–590, which are arranged in 13 transmembrane helices (TMs 1–13) and 3 cytoplasmic helices (HC1–HC3), with N- and C-termini located at extracellular and cytoplasmic sides, respectively (Fig. 1 and Supplementary Fig. 5a). Most extracellular loops (ELs) and intercellular loops (ILs) are well resolved, including a short helix connecting TM3 and TM4 (IL2a). Missing residues occur only in regions either N- or C-terminal to the final model, presumably due to conformational flexibility. Topology of NHE1 determined in the current study is mostly consistent with a previous prevalent model assessed by substituted cysteine accessibility analysis[34]. One major discrepancy is, however, the existence of a TM helix in the unresolved N-terminal region, which is predicted to be a cleavable signal peptide using the SignalP-5.0 server[35], in line with a concept that a large extracellular N-terminal generally harbors a cleavable signal peptide[19,36]. Another minor discrepancy is the reentrant loop between TM9 and TM10 in the Wakabayashi's hypothetical model, which is actually arranged into two TM helices (TM9–10) in our structure. Similar to previously reported prokaryotic homologs[16–18], the TMD structure of NHE1 is organized into a homodimer. Each monomer is made up of a dimerization domain (TMs 1–3 and 7–10) and a core domain (TMs 4–6 and 11–13; Figs. 1c, d). The cross-section of the NHE1 dimer has a rectangular shape that is ~84-Å long

and 48-Å wide. The nanodisc dimension is ~109 × 129 Å, which provides sufficient lipid surface area to accommodate the complex. The central symmetry axis passes through the dimerization domain, which is ~37 Å in height, i.e., ~10-Å thinner than that of the core domains on both sides. In addition, the overall folding of each NHE1 subunit shows a pseudo 2-fold symmetry, with the symmetry axis parallel to the membrane plane, which relates TMs 1–6 to TMs 8–13. In particular, both TM5 and TM12 are characteristically unwound in the middle, resulting in fragments TM5a, TM5b, TM12a, and TM12b, and the two unwound peptides cross each other in the vicinity of the pseudo 2-fold axis. These helix breaks are likely to participate in formation of the ion permeation pathway.

The observable N-terminal EL1 (i.e. residues 87–99) extends from one subunit and resides above TMD of the other subunit, and it has not been seen in any dimeric structure of prokaryotic homologs and horse NHE9. As mentioned above, three cytoplasmic helices, HC1 (residues 518–538), HC2 (residues 543–562), and HC3 (residues 570–590) were determined in the NHE1-CHP1[K/cariporide] complex structure (Fig. 1c, d). Consistent with a previous report[11], HC1 is a juxtamembrane helix embedded in a surface cleft of CHP1. It is followed by the amphipathic helix, HC2, adjacent to the intracellular ends of TMs 4, 6, and 9 of NHE1 and presumably interacting with the membrane surface (Fig. 1c). In the NHE1-CHP1[Na/6.5] complex, HC1 and HC2 helices are perpendicular to each other. Moreover, the two CHP1 molecules reside at opposite ends of the rectangular-shaped NHE1 dimer, by interacting with IL6, HC1, and HC2, resulting in an ~30° angle between the CHP1 molecule and the membrane plane. Strikingly, in the context of the NHE1-CHP1[K/cariporide] complex, distal ends of the two CHP1 subunits move close to each other and towards the membrane plane, as evidenced by observations that the height of the E-helix of the fourth EF hand (E[4th] helix) of CHP1 relative to the membrane is reduced by 11 Å and that the end-to-end distance between E[4th] helices becomes 23-Å shorter than that in the NHE1-CHP1[Na/6.5] complex (Fig. 1a, b). Consequently, both HC1 and CHP1 become nearly parallel to the membrane plane and thus the angle between HC1 and HC2 helices reduces from 88° to 63°. Other NHE1 structural elements, such as IL2a and HC3, also interact with CHP1. For instance, the HC3 helix is now located underneath the NHE1 dimerization domain around the 2-fold symmetry axis and is fixed in between the CHP1 subunit and NHE1 dimer. Distance between the helix axes of the antiparallel HC3 pair is 9.5 Å, in agreement with a previous report showing that residues 560–580 play a pivotal role in dimerization of the cytoplasmic tails and are thus crucial for both NHE activity and $H^+$ sensing[37]. We hypothesize that absence of the HC3 helices in the NHE1-CHP1[Na/6.5] complex model is caused by the splay-opened CHP1, thus releasing the HC3 helices from the TMD dimer.

Superposition of the complex structures determined under different conditions demonstrates that the NHE1 dimer structures are nearly identical in the presence of $Na^+$ ions (Supplementary Fig. 5b), at either pH 7.5 or pH 6.5, with a root-mean-square deviation (RMSD) of 0.6 Å for 840 Cα-atom pairs. This pH-independency is distinct from observations in PaNhaP showing more prominent conformational changes upon altering pH, with an RMSD of 1.6 Å for 811 Cα-atom pairs between dimers at pH 8 (PDB ID: 4CZ8) and pH 4 (PDB ID: 4CZ9)[17]. However, structural comparison between the NHE1-CHP1[K/cariporide] and NHE1-CHP1[Na/6.5] complexes indicates that the protomer structure undergoes a conformational change, with an RMSD of 2.3 Å for 420 Cα-atom pairs between the corresponding two protomers. Such a conformational change within the protomer seems to be essential for the formation of a

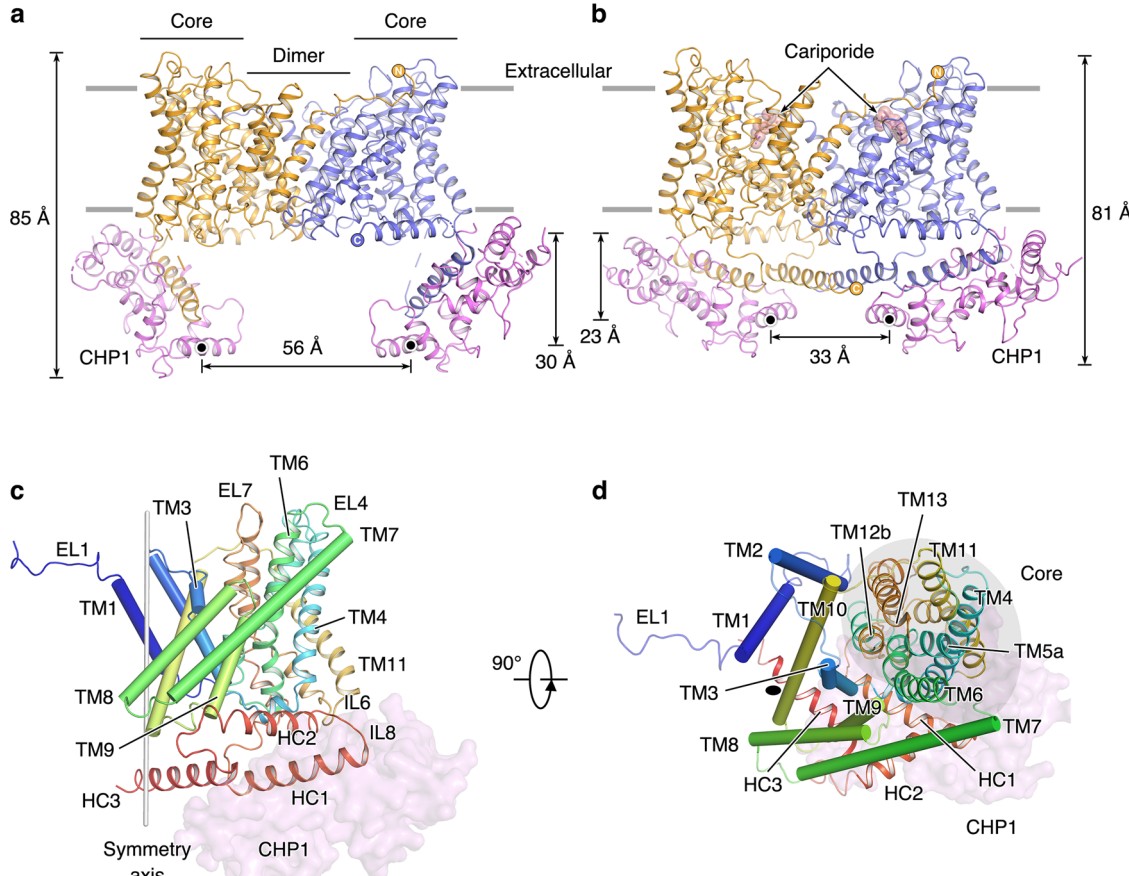

**Fig. 1 Architecture of the NHE1-CHP1 complex. a, b** Overall structures of the NHE1-CHP1 complex, in the absence and presence of cariporide, respectively. Two subunits of NHE1 are colored in orange and blue, and two CHP1 subunits are colored in magenta. Gray lines represent boundaries of the cell membrane. The black dots depict the centers of mass (COMs) of the E$^{4th}$ helix of CHP1. The height of the complex and the distance between COMs are indicated. **c, d** Structure of the NHE1 protomer in the NHE1-CHP1$^{K/cariporide}$ complex, viewed in the membrane plane and from the extracellular side, respectively. The peptide backbone of NHE1 is colored in a rainbow scheme, with blue and red for the amino and carboxyl termini, respectively. The core and dimerization domains are shown in cartoon and cylinder, respectively. The CHP1 molecule is displayed as a transparent pink surface model. 2-fold symmetry axis are depicted as a gray stick or black oval. The core domain is highlighted with a gray oval.

cariporide binding pocket in the extracellular portion of the TMD (Supplementary Fig. 5c), and thus the cariporide-bound NHE1-CHP1 complex is in a distinct conformational state from the other two cariporide-free complexes. Moreover, structural comparison of the NHE1 TMD of the NHE1-CHP1$^{Na/6.5}$ complex with horse NHE9 (PDB ID: 6Z3Y) and PaNhaP (PDB ID: 4CZ8) indicates that they adopt a similar overall shape and folding topology featuring highly tilted TM helices (Supplementary Fig. 5d). Superimposition of the NHE1 structure with horse NHE9 and PaNhaP yields RMSDs of ~2.2 Å for the horse NHE9 dimer (662 Cα atom-pairs), ~2.1 Å for the horse NHE9 protomer (324 Cα atom-pairs), ~2.6 Å for the PaNhaP dimer (664 Cα atom-pairs), and ~2.3 Å for the PaNhaP protomer (337 Cα atom-pairs), respectively, suggesting that the structural difference is mainly derived from the NHE protomer.

**Dimer interface of the NHE1-CHP1 complex.** In the complex structure, four regions from each subunit contribute to NHE1 dimerization, including EL1, extracellular and intracellular sides of the dimerization domain and HC3 (Fig. 2a). In particular, unlike the prokaryotic homolog transporters, the N-terminal EL1 of one NHE1 subunit resides on TMs 8 and 9 and interacts with EL5 and the extracellular parts of TMs 6 and 7 from the other subunit (Fig. 2b). Inside the membrane region, the dimerization of NHE1 is largely mediated by extensive van der Waals interactions

between TMs 1, 7, 8, and 10. Most involved residues from these helices are hydrophobic. Residues from one subunit, such as V99, F103, S106, L107, I109, L110, L114, I117, and V121 on TM1, interact with I326, I329, L332, F333, L336, Y337, M340, L343, and L347 on TM8′ from the other subunit (the prime depicts being from the symmetry mate) (Fig. 2d). The N-terminus of TM8, which is delimited by P331 and its preceding IL4, is closely packed with the intracellular side of TM10 helices from both subunits as well as TM1′. Residues T378, Y381, and M385 on both TM10 helices are located in the close vicinity of the 2-fold symmetry axis, and directly contribute to the dimerization interface (Fig. 2d). Moreover, also on the cytoplasmic side, the dimeric HC3 helices are positioned around the 2-fold symmetry axis and interact with each other by hydrophobic interactions involving M582, A585, I586, and V589 (Fig. 2e). This homotypic interaction is important for dimerization of the cytoplasmic tails of NHE1. The total buried accessible-surface area (ASA) upon dimerization is ~6700 Å$^2$ and ~7300 Å$^2$ for the NHE1-CHP1$^{Na/6.5}$ and NHE1-CHP1$^{K/cariporide}$ complexes, respectively.

A cavity of ~3100 Å$^3$ in size was visualized on the extracellular side of the central part of the dimer, surrounded by TMs 1, 2, 7, 8, and 9 from the two dimerization domains (Fig. 2c). A number of hydrophobic residues, including aromatic F164, Y339, F382, W386, and F397, form the wall of this hydrophobic cavity. Six strip-shaped EM densities are located inside this cavity and

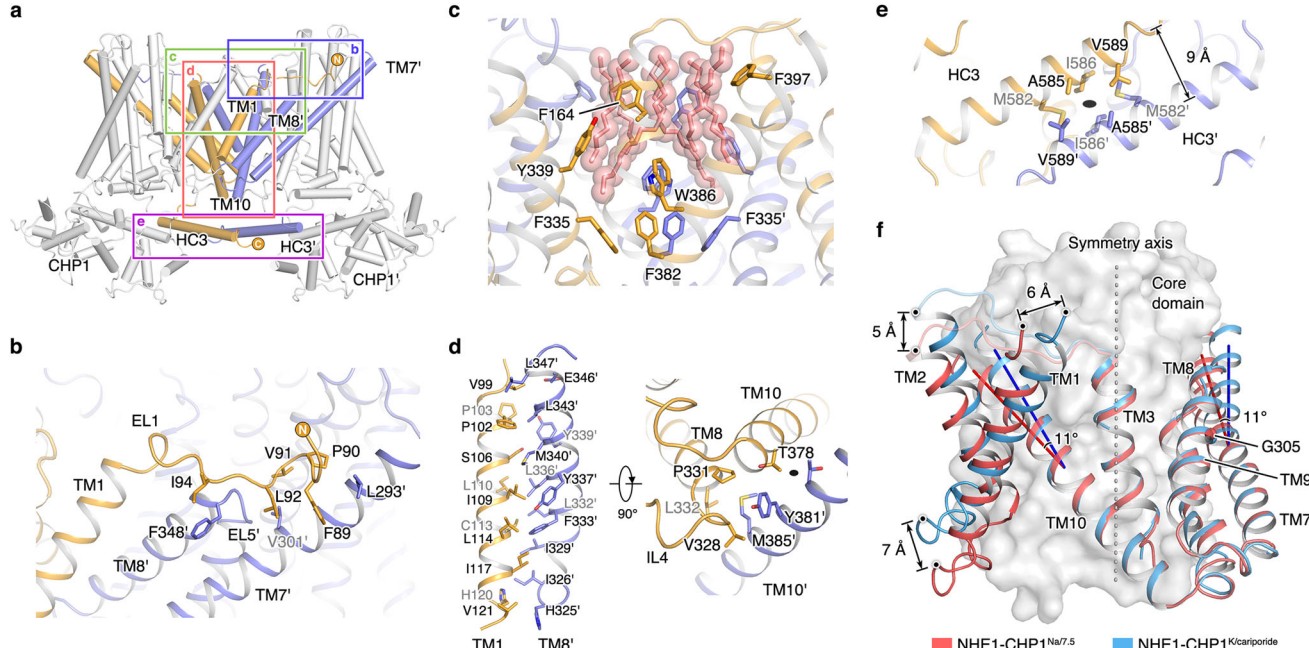

**Fig. 2 Dimerization interface of the NHE1-CHP1 complex. a** Helices from the dimerization domains in the NHE1-CHP1$^{K/cariporide}$ complex shown in cartoon. Four regions involved in dimerization are boxed. **b** Interactions between EL1 and TM7' or TM8' in the extracellular side. Hydrophobic residues are shown as sticks. **c** Hydrophobic cavity between the two dimerization domains facing the extracellular side. Aromatic amino acid residues and putative lipids (salmon) within the cavity are shown as sticks. **d** Hydrophobic interactions between TMs 1, 8, and 10 from one protomer and TMs 8' and 10' from another protomer. Sidechains of residues in dimer contact are showed in sticks. The 2-fold symmetry axis is depicted as a black oval. **e** Dimerization of the HC3 helices around the 2-fold axis. **f** Structural comparison of the dimerization domain between NHE1-CHP1$^{K/cariporide}$ complex (in blue) and the NHE1-CHP1$^{Na/7.5}$ complex (in red). The core domain is depicted as a gray surface model. The displacements of helix and helix bending are indicated. The Cα atom of G305 is shown as a sphere. The 2-fold symmetric axis is depicted as a dash line.

remain nearly identical in all three types of complexes (Fig. 2c and Supplementary Figs. 2e, 3f, and 4e). These densities are likely to represent hydrophobic tails of lipid molecules, similar to the homolog structure of PaNhaP (PDB ID: 4CZ8) in which two tails from one lipid molecule were found to be located inside the corresponding hydrophobic pocket. Since an empty huge cavity in a membrane protein would be energetically unfavorable in the membrane environment, we speculate that the observed lipid molecules inside the cavity are required for the stability of the NHE1 dimer.

Structural comparison of the NHE1 dimer in the NHE1-CHP1$^{Na/6.5}$ form with that of the cariporide-bound inhibited state showed little conformational difference in the dimer-interface region composed of TMs 3, 7, 8, 9, and 10, supporting the notion that these helices play a major role in NHE1 dimerization. The RMSD of the five TM helices is ~1.1 Å between the two states. Although it does not directly contribute to the dimerization interface, TM3 is maintained nearly identical in the two states; in fact, TM3 is fixed by extensively interacting with adjacent TM helices, including TMs 8, 9, and 10. In contrast, TMs 1 and 2 appeared to be of greater mobility and were thus excluded from the two-state analysis. Interestingly, the two tilted long helices, TM7 and TM10, which connect to helices of the dimerization domain with those from the core domain, show ~50° or ~40° off from the 2-fold symmetry axis, respectively; their extracellular halves both bend ~11° between the two states (Fig. 2f). This flexibility is likely to be important for the relative conformational change between the core and dimerization domains during the transport cycle. Previously, a G305R mutation was reported to be associated with the ataxia-deafness Lichtenstein-Knorr syndrome and to retain the NHE1 molecules at the endoplasmic reticulum[13]. In our structure, G305 is located at the middle of TM7, facing both TMs 8 and 9. Substitution of this Gly residue with Arg would disrupt the tightly packed 7-8-9 helix bundle (Supplementary Fig. 5f), thus hindering dimerization of NHE1. In addition, such a mutation would make the backbone of TM7 more rigid, probably hampering the conformational transition associated with transport activity.

**Inward-facing conformational state of the NHE1-CHP1 complex.** We determined structures of the NHE1-CHP1 complex in the presence of Na$^+$ under two different pH conditions. Little NHE1 conformational change was visualized between these two states (Supplementary Fig. 5b). A large funnel between the dimerization domain and the core domain in each protomer is accessible only from the intracellular side, indicating that in both pH conditions the NHE1-CHP1 complex adopts an inward-facing conformation. The funnel is ~23-Å deep and is formed by TMs 1, 2, 3, 5, 6, and 10 (Fig. 3a). A cluster of proton-titratable residues are located inside of the funnel chamber, including E131 in TM2 (E131$^{TM2}$), D172$^{TM3}$, D238$^{TM5}$, D267$^{TM6}$, and E391$^{TM10}$, resulting in a highly negatively charged cavity which is suitable for both cation binding and proton sensing. According to previous mechanistic models of NHE[18], at the inward-facing conformation, the Na$^+$ ion would bind to the transporter at higher pH but be displaced by a proton in a lower pH environment. Nevertheless, we did not observe a bound ion in the NHE1-CHP1 complex, possibly due to the moderate resolution of the EM maps. To explore ion binding site(s) of NHE1, we overlaid the NHE1 structure onto a Tl$^+$-bound PaNhaP structure (PDB ID: 4CZA) around the cation binding site, resulting in super-imposable TMs 3, 5, 6, and 12 (Fig. 3b). Upon the structural superposition, it is apparent that the cation binding site of PaNhaP is conserved in NHE1. A putative substrate ion is likely to be directly ligated in NHE1 by five oxygen atoms from

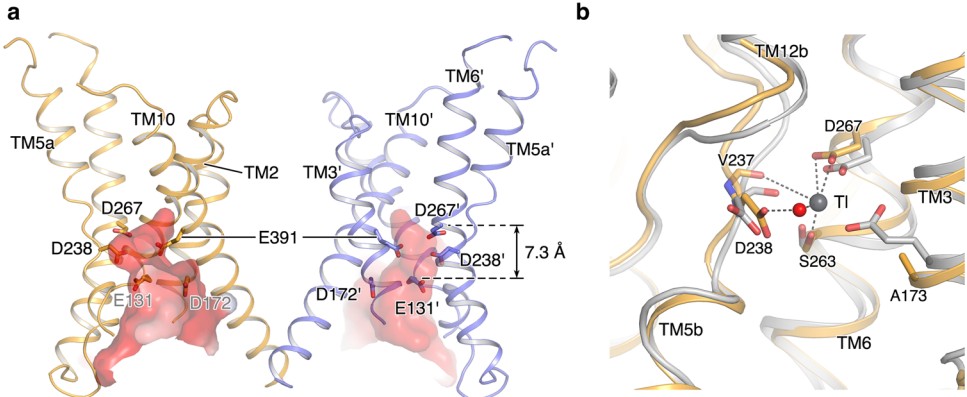

**Fig. 3 Putative cation binding site of the inward-facing NHE1-CHP1$^{Na/7.5}$. a** The intracellular funnel between the dimerization domain and core domain is displayed in electrostatic surface. Proton-titratable residues in this funnel are displayed in sticks. **b** Putative cation binding site of the inward-facing NHE1-CHP1$^{Na/7.5}$ complex. NHE1-CHP1$^{Na/7.5}$ complex and PaNhaP structures are colored in orange and gray, respectively. Thallium ion and a water molecule in PaNhaP are shown in gray and red spheres, respectively. TMs and putative cation binding residues of the NHE1-CHP1$^{Na/7.5}$ complex are indicated.

sidechains of S263$^{TM6}$, D267$^{TM6}$, mainchain of V237$^{TM5}$, and a water molecule. Among these coordination groups, the exact identity of the polar sidechain of S263$^{TM6}$ appear critical for ion transport[18]. D267$^{TM6}$ is directly involved in cation binding, consistent with previous data showing that the D267N mutant abolishes transport activity, whereas D267E retains the activity[38]. In addition, D238$^{TM5}$ is indirectly involved in cation binding by mediating the direct coordination of the water molecule, in line with a previous report showing that the D238N mutation does not affect the activity[38]. In addition, E73 is found to take part in ion binding in the PaNhaP structure, yet is not essential for the transport activity[18]. The equivalent position in NHE1 becomes A173$^{TM3}$.

Besides D267$^{TM6}$ and D238$^{TM5}$ participating in the substrate binding, more titratable residues that we found inside the funnel have been postulated to act as allosteric modulation sites in NHE1 to sense intracellular pH (refs. [39,40]). For instance, E391$^{TM10}$ from the dimerization domain, located at the membrane level similar to D267$^{TM6}$, is functionally important, evidenced by the E391Q mutant showing reduced activity, whereas E391D retains the activity[38]. However, E391$^{TM10}$ is positioned ~13-Å away from the putative cation binding site and thus is unlikely to directly participate in substrate binding. A trypsinolysis experiment suggested that the E391Q mutant is more sensitive to trypsin digestion in an in vitro assay than the WT NHE1, especially in the presence of Na$^+$ ions[38]. We thus speculate that E391 is critical for the folding and/or stability of NHE1, particularly in its non-protonated state. Another acidic residue, D172$^{TM3}$ is located on the cytoplasmic side of E391;$^{TM10}$ the Cα-Cα distance between D172 and E391 is ~11 Å. In contrast to E391, mutants D172E/N/Q do not hamper NHE1 activity[41]. In addition, around the cation binding site, E131$^{TM2}$ is positioned on the side opposite to D172$^{TM3}$ and is 7-Å below D267$^{TM6}$, thus being more exposed to the cytoplasmic environment. According to a previous report, the E131D mutant showed no effect on the exchange activity, and E131Q resulted in a slight acidic-shift of the pH dependency. In contrast, charge reversal mutants E131R/K resulted in a large acidic-shift of pH dependency, presumably by over-reducing the p$K_a$ value of the orthosteric H$^+$-binding site; their Hill coefficients were also reduced from 1.4 (WT) to ~1.0 (ref. [42]). Based on the current structure of NHE1, we propose that E131$^{TM2}$ near the cation binding site acts as a pH sensor, and upon protonation this acidic residue partly contributes to transport cooperativity by means of accelerating substrate cation release.

**Inhibitor-binding pocket of the NHE1-CHP1 complex**. NHE1 inhibitors are currently categorized into several different classes, the most prominent being the pyrazinoylguanidine and benzoylguanidine derivatives[43]. Whereas the pyrazinoylguanidine-type inhibitors are derivatives of amiloride, the benzoylguanidine-type inhibitors replace the pyrazine core with a phenyl ring and usually show higher NHE1 specificity[44]. To gain insight into the inhibitory mechanism, we determined the complex structure of NHE1-CHP1 bound with cariporide, a potent benzoylguanidine-type inhibitor with an IC$_{50}$ of ~30 nM[45]. Cariporide bound from the extracellular side to a negatively charged pocket in each NHE1 subunit. This pocket is located in between the dimerization domain and core domain, ~10-Å in depth, and surrounded by a bundle of helices, namely TMs 3, 6, 8, 9, and 12 (Fig. 4). Strikingly, the EL1 loop from the opposing subunit resides above TMs 8 and 9, and this interaction is probably essential for the integrity of the pocket. Such structural complementation are distinct from the homologs PaNhaP, MjNhaP, and horse NHE9; particularly the corresponding N-terminal 15 residues of MjNhaP which are dispensable for its activity[18]. In the context of the NHE1-CHP1$^{K/}$ $^{cariporide}$ complex, D267$^{TM6}$, which is strictly conserved among NHE-like transporters and a critical component of the substrate ion binding pocket in the inward-facing conformation, now points upward from the bottom of the inhibitor-binding pocket and would be accessible from the extracellular side if the inhibitor were absent (Fig. 4a and Supplementary Fig. 6). D267$^{TM6}$ directly interacts with the positively charged guanidine group of cariporide. This structural observation indicates that the cariporide-bound complex is stabilized in the outward-facing state. In addition, F162$^{TM3}$ is located in a helical bulge which seems to be stabilized by two conserved proline residues at positions 167 and 168 (Fig. 4b, c and Supplementary Fig. 6). The sidechain of this F162 residue interacts with both the guanidine group and phenyl ring of cariporide by cation−π and π−π interactions in a parallel and T-shaped configuration, respectively. This structural observation is consistent with a previous report that the F162S mutation dramatically increases $K_i$ of cariporide[46]. In addition to D267$^{TM6}$ and F162$^{TM3}$, the guanidine group of cariporide is also coordinated by the sidechain of E346$^{TM8}$, in harmony with previous studies[47,48]. Considering that the guanidine group of cariporide is of a similar size and charge with a (partially) hydrated Na$^+$ ion, we thus hypothesize that F162$^{TM3}$, D267$^{TM6}$, and E346$^{TM8}$ also coordinate the Na$^+$ ion in the outward-facing state during ion exchange, evidenced by mutations at these sites hampering both inhibitor binding and transport activity[49]. Since

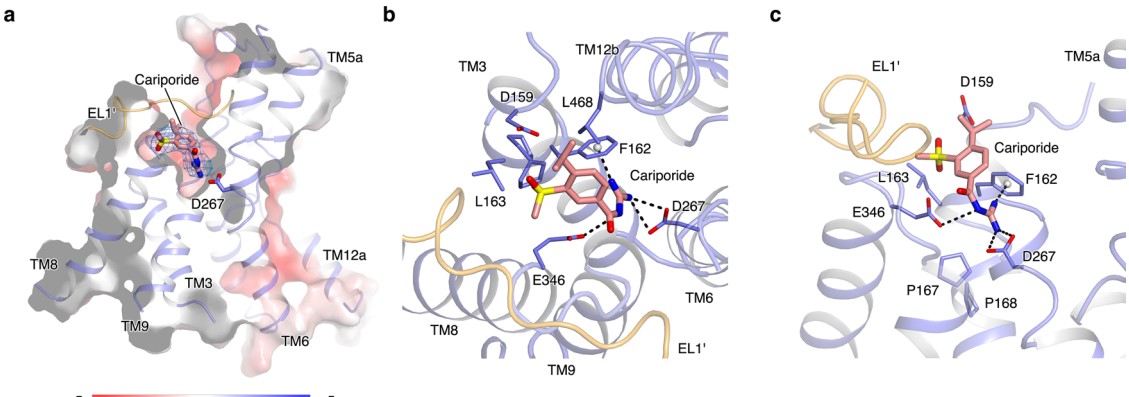

**Fig. 4 Cariporide binding pocket of the NHE1-CHP1$^{K/cariporide}$ complex. a**. Slice-through electrostatic surface of the NHE1-CHP1$^{K/cariporide}$ complex. The cariporide molecule is displayed in sticks, and its corresponding EM density is shown in mesh. **b**, **c** Binding site of the inhibitor cariporide. The cariporide molecule (salmon) and its surrounding residues are shown as stick models. Two subunits of NHE1 are colored in orange and blue.

it takes part in both the inward- and outward-facing substrate-binding sites, the D267$^{TM6}$ residue is most likely a critical residue for the alternating-access mechanism of cation translocation.

Besides the guanidine group, the cariporide molecule harbors a methylsulfonyl group and an isopropyl group at the meta- and para-position of the phenyl ring, respectively. In the NHE1-CHP1$^{K/cariporide}$ complex structure, the methylsulfonyl group is buried within a sub-pocket formed by D159, L163, D95′, H98′, and V99′, consistent with a previous observation that small moieties commonly occupy this position in most NHE1 specific inhibitors. In particular, mutations of L163 reduce inhibitor sensitivity in NHE1 (ref. [50]). Because of a phenylalanine substitution at the equivalent position, NHE3 is less sensitive to those NHE1 inhibitors[48]. In contrast to the methylsulfonyl group, the isopropyl group attached to the para-position in cariporide is accessible from the extracellular space. Since the residues involved in cariporide binding are from both the core and dimerization domains, we speculate that cariporide inhibits the NHE1 activity mainly by blocking the relative sliding movement between the two domains in addition to competing with Na$^+$ for the extracellular substrate-binding site.

**Conformational change during transport cycle.** To elucidate conformational change in the NHE1 protomer upon transition from the inward- to outward-facing state, the NHE1-CHP1$^{Na/6.5}$ structure was overlaid on the NHE1-CHP1$^{K/cariporide}$ structure based on their dimerization domains. It turns out that the intracellular side of the core domain moves ~5 Å towards the extracellular side, defined by displacement of the center of mass (COM) of the residue group of L186, V245, E253, F439, P446, and V503. On the extracellular side, a similar 5-Å COM displacement was also observed; however, it was not perpendicular to the membrane plane and yielded an ~10° rotation of the core domain. Consequently, the extracellular halves of TMs 7 and 10, i.e., the connecting helices from the dimerization domain, exhibit the abovementioned tilting between the two states (Fig. 2f). Of particular interest, upon the outward- to inward-facing transition, the Cα atom of D267$^{TM6}$ is displaced by 5 Å towards the extracellular side, and its sidechain rotates 24°. A potential steric hindrance would occur between D267$^{TM6}$ of the inward-facing state and P239$^{TM5}$ of the outward-facing state, implying that the ion binding site shown in the inward-facing state is incompatible with the cariporide bound outward-facing state (Fig. 5b). Taken together, we envision that the NHE1-CHP1 complex implements an elevator-like transport mechanism similar to that proposed previously for electrogenic bacterial Na$^+$/H$^+$ antiporters[16,20], in

which the dimerization domain is fixed and the core domain undergoes up and down movement across the membrane.

The core domain itself does not show discernible internal conformational change between the inward- and outward-facing states, supported by a RMSD of 1.3 Å for 177 Cα-atom pairs between the corresponding models (Supplementary Fig. 7a, b). Moreover, mutation of the conserved E262$^{TM6}$ substantially decreases NHE1 transport activity[51]. Whereas it is buried inside the core domain forming a salt bridge with R425$^{TM11}$, E262$^{TM6}$ is not accessible from either side of the membrane (Supplementary Figs. 6 and 7a, b), suggesting that E262$^{TM6}$ does not directly participate in ion coordination. Nevertheless, further investigation is required for better understanding of its functional role. Furthermore, although the main part of the core domain functions as a rigid body during the state transition, TM5b and its following loop IL3 exhibit a dramatic local conformational change up to 10 Å. In the inward-facing conformation, the hydrophobic middle part of TM5 becomes fully stretched, and TM5b is arranged to form a short 3$_{10}$ helix, made up of $^{239}$PVAVLA$^{244}$ (Supplementary Fig. 7c–f). A salt-bridge bond, R500−E247, is supposedly to stabilize this peculiar conformation. In contrast, in the outward-facing state, TM5b is rearranged to an α-helix by incorporating an addition part from IL3$^{245}$VFEEI$^{249}$, and this new TM5b is lifted by a charge-dipole interaction between D238$^{TM5}$ and the N-cap of TM12b (Supplementary Fig. 7d–f). This alternative conformation is accompanied by switching from the R500−E247 interaction to a new salt-bridge bond between R500 and E248. In agreement with the structural observation, interruption of the N-cap interaction in the D238A mutant abolished transport activity without affecting protein expression or plasma membrane localization (Fig. 5f and Supplementary Fig. 8a, b). Previous site-directed mutagenesis of E247, E248, and R500 suggested that these residues play critical roles in the transport process[52,53].

In addition to the relative movements of the core and dimerization domains, another hallmark conformational change between the two functional states is the movement of CHP1. In the context of inward-facing NHE1-CHP1$^{Na/7.5}$ and NHE1-CHP1$^{Na/6.5}$ complexes, the CHP1 molecule exhibits high mobility relative to the NHE1 dimer, resulting in featureless CHP1 density in the EM maps. Two distinct NHE1-CHP1 configurations are consistently observed in the inward-facing complexes. One of them shows splay-opened CHP1 which forms an ~35° angle to the membrane plane (Supplementary Fig. 9a–c). In the other configuration, the CHP1 molecule moves close to the central axis and undergoes a clockwise rotation relatively to the first

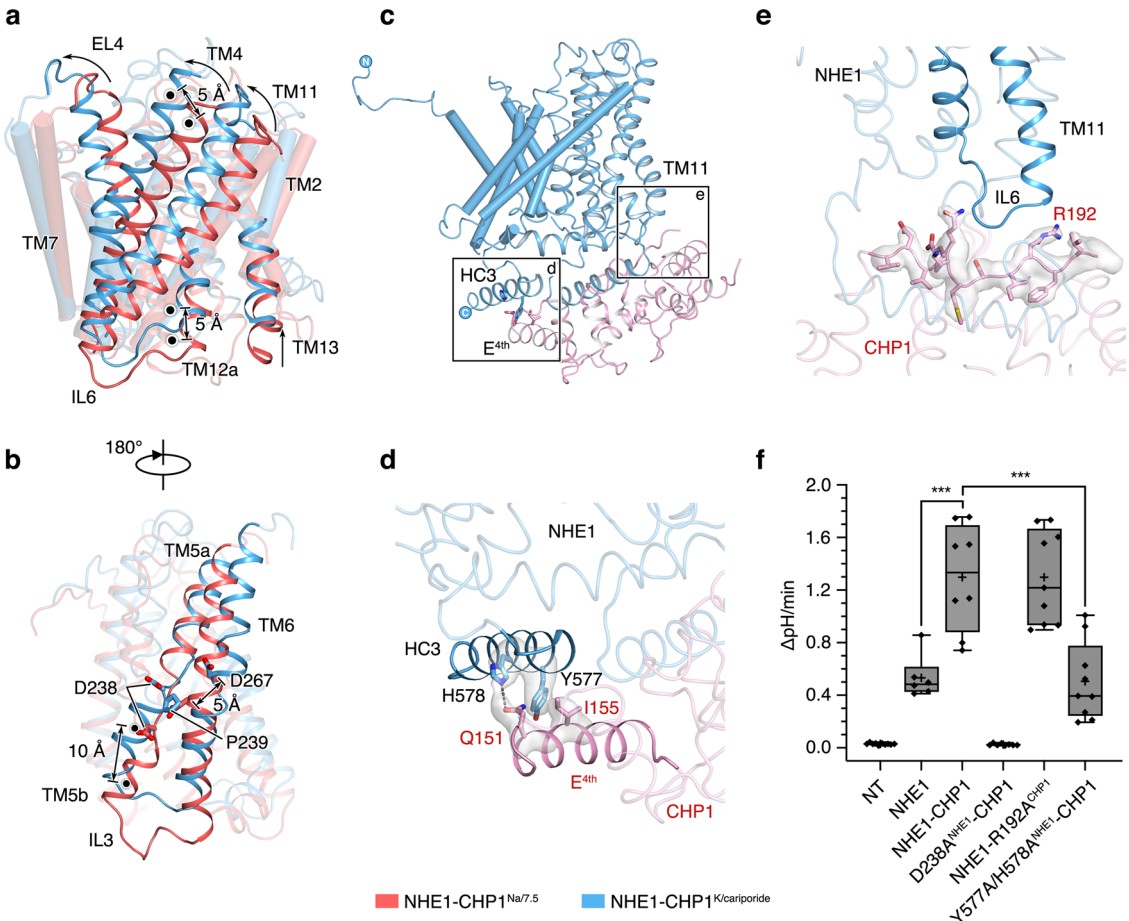

**Fig. 5 Conformational change between the inward- and outward-facing NHE1. a, b** Conformational change of the core domain relative to the dimerization domain, between the inward- (red) and outward-facing (blue) states. Dimerization domains are used for superimposition and shown in cylindrical helices. The COMs of residues of the core domain at extracellular side (V214, L224, A281, W411, L468, and M476) and intracellular side (L186, V245, E253, F439, P446, and V503) are depicted as black dots. Distances between COMs are indicated. Panel **a** and **b** differ by 180°. **c–e** Interaction between NHE1 and CHP1. The NHE1 and CHP1 molecules are colored in blue and pink, respectively. Sidechains are depicted as sticks and the cryo-EM density map is shown in surface models. **f** Measurement of NHE1 activity. AP-1 cells were transiently transfected with mCherry fluorescent protein alone (NT, no transporter), wild-type NHE1 tagged at its C-terminus with mCherry alone, or wild-type or mutant NHE1 co-expressed with either wild-type CHP1 or mutant CHP1 (R192A$^{CHP1}$), as indicated. The mCherry signal was used to identify transfected cells. The cells were acid-loaded by prepulsing with $NH_4^+$ (as described in Methods section) and NHE1 activity was defined as the initial linear rate of pH recovery (ΔpH/min) upon addition of $Na^+$ to the bathing medium. Data are plotted as a box plot, with the box encompassing the interquartile range (25th−75th percentile) including the mean (plus sign) and the median (straight line), and the whiskers showing the 99% confidence interval. Each symbol represents a single cell, and between 6 and 10 cells (n value) were analyzed from two independent experiments (NT, n = 10; NHE1, n = 6; NHE1-CHP1, n = 8; D238A$^{NHE1}$-CHP1, n = 8; NHE1-R192A$^{CHP1}$, n = 9; Y577A/H578A$^{NHE1}$-CHP1, n = 9). Significance from WT was determined by one-way ANOVA (total degrees of freedom = 49; F value = 42.1, P value = 1.094 × 10$^{-15}$), with a post-hoc Tukey test (***, NHE1:NHE1-CHP1, P = 3.42096 × 10$^{-5}$, NHE1-CHP1: Y577A/H578A$^{NHE1}$-CHP1, P = 2.16504 × 10$^{-6}$). Source data are provided as a Source Data file.

conformation, as viewed from the cytoplasmic side (Supplementary Fig. 9b–d). We hypothesize that flexibility of the CHP1 molecules is important for pH-sensing, and it is likely that other regulatory partners, like PIP$_2$, will further bind to the NHE1-CHP1 complex and restrict the conformational heterogeneity of CHP1. The connection between CHP1 and NHE1 is mainly mediated by IL8, a short hinge loop made up of residues $^{538}$CGHYG$^{542}$ between HC1 and HC2 (Fig. 1c). Whereas HC1 is embedded in CHP1, HC2 assumes more extensive interactions with TMD of NHE1. Therefore, mobility of the attached CHP1 molecule is likely to inversely correlate with the rigidity of the hinge in IL8. Moreover, the surfaces of CHP1 and NHE1 molecules display complementary patches of negative and positive electrostatic potential around their interactive regions (Supplementary Fig. 9e−g), and the associated interactions are likely to be subjected to regulation by intracellular pH.

Interestingly, the CHP1 molecule favors a well-defined conformation in the outward-facing NHE1-CHP1$^{K/cariporide}$ complex, in sharp contrast with the high heterogeneity of CHP1 conformations observed in the inward-facing state. Structural features of the mainchain and bulky sidechains of CHP1 become better defined in the inhibitor-bound structure due to stabilization of CHP1 by more interactions with the NHE1 TM domain. For instance, the sidechains of Q151$^{CHP1}$ and I155$^{CHP1}$ form a hydrogen bond and hydrophobic interaction with H578$^{NHE1}$ and Y577$^{NHE1}$, respectively; and R192$^{CHP1}$ forms a charge-dipole interaction with the C-cap of TM11 (Fig. 5d, e). The IL7 loop is rich of basic residues ($^{509}$KKKQETKR$^{516}$), and has been proposed to bind with the phosphate head groups of PIP$_2$ (ref. [54]), which was not supplemented in our sample preparation. Intriguingly, IL7 was resolved only in the outward-facing state, being positioned underneath the TM domain and

contacting with IL3 (Supplementary Fig. 7f). Thus, this stabilized IL7 loop in the outward-facing NHE1-CHP1$^{K/cariporide}$ complex appears to be inaccessible by the lipid bilayer including PIP$_2$. More specifically, upon the NHE1 transition to the outward-facing conformation, charged residues are rearranged, generating a more negative electrostatic potential on the cytoplasmic surface of TMD. This newly formed surface patch attracts the positively charged IL7 presumably away from interacting with the PIP$_2$ molecules (Supplementary Fig. 9e, f). The CHP1 molecule, in turn, becomes stabilized in the vicinity of NHE1 TMD. So far, residues on IL7, IL8, and HC2 have been identified to be critical for PIP$_2$ modulation[54,55], which are all close to the CHP1 binding site. Hence, we proposed that PIP$_2$ modulates the activity of NHE1 by affecting CHP1 conformation, particularly for the CHP1 in the inward-facing conformation which exhibits highly motility relative to NHE1 in the absence of PIP$_2$. The structure of a PIP$_2$-bound NHE1-CHP1 complex is required to fully understand the modulation mechanism of PIP$_2$ on NHE1 transport activity.

To study how the CHP1 binding regulates NHE1 activity, we interrupted the interactions between CHP1 and NHE1 observed in the cariporide-bound outward-facing conformation by introducing mutations of R192A$^{CHP1}$ and Y577A/H578A$^{NHE1}$ and carried out AP-1 cell-based transport assay[56]. The WT CHP1 profoundly facilitates NHE1 accumulation at the cell surface and enhances NHE1 transport activity (Fig. 5f and Supplementary Fig. 8a−d). The R192A$^{CHP1}$ does not interrupt the transport activity (Fig. 5f). Despite an expression level comparable to the WT complex, the activity of the double mutant Y577A/H578A$^{NHE1}$ decreased by 50% relative to the WT complex, indicating that the CHP1−HC3 interaction is critical for the ion exchange (Fig. 5f and Supplementary Fig. 8a−c). Considering that these two sites are far away from the cation binding site and are flexible in the inward-facing conformation, we speculate that this CHP1−HC3 interaction is important for conformational transition from the inward-facing state to the outward-facing state. However, to fully interpret how CHP1 enhance the NHE1 activity, a more complete complex, in the presence of PIP2 or other regulatory proteins, seems to be required.

Here we elucidated the structures of the human NHE1-CHP1 complex in both the inward-facing and cariporide bound outward-facing conformational states in the lipid bilayer. In the context of the inward-facing complexes, a large negatively charged funnel located between the dimerization domain and core domain, where the acidic residue D267 is critical for the cation binding at the floor of the pocket. In the cariporide-bound outward-facing state, the cariporide molecule competes with Na$^+$ for binding with D267 from the extracellular side of the NHE1. By comparing these two conformational states, we conclude that the NHE1 undergoes an elevator-like movement to exchange ions and that D267 is the residue key for binding Na$^+$ and H$^+$ substrates. Furthermore, our structures show that the CHP1 molecules differentially associate with the two major states of the NHE1 dimer, and such difference is likely to be important for the regulation of the NHE1 activity by CHP1.

## Methods

**Expression and purification of human NHE1-CHP1 complex.** The cDNAs of the human NHE1 (UniProtKB accession: P19634) and human CHP1 (UniProtKB accession: Q99653) genes were amplified from an HEK293 cDNA library. The NHE1 gene was subcloned into a modified pEG BacMam vector that encodes a PreScission Protease (PPase) recognition site followed by a superfolder GFP (sfGFP) and a Twin-Strep tag at the C-terminus of the product (Supplementary Table. 2). The CHP1 gene was similarly cloned into a modified pEG BacMam vector, with a PPase cleavage site, 8xHis and FLAG tags being fused to the C-terminus of the gene product. Subsequently, one copy of the NHE1 gene, two copies of the CHP1 gene (to enhance its expression level), and associated

expression elements were subcloned into a pBIG1a vector-based expression cassette which was used to co-express recombinant complexes. Recombinant proteins were expressed in HEK293F cells using the Bac-to-Bac baculovirus expression system (Invitrogen, USA). Specifically, the baculovirus was generated in Sf9 insect cells, and P2 viruses were used to infect HEK293F cells. The cells were grown at 37 °C in suspension supplemented with 1% (v/v) fetal bovine serum and 5% CO$_2$ in a shaking incubator. After 12 h, sodium butyrate (10 mM final) was added into the culture and cells were incubated for another 48 h before harvesting.

HEK293F cells expressing the NHE1-CHP1 complex were collected and resuspended in a purification buffer A (20 mM HEPES pH 7.5 or 6.5, 150 mM NaCl, 5 mM β-mercaptoethanol (β-ME), 500 nM CaCl$_2$, and protease inhibitor cocktail (Roche, Swiss)). The membrane was extracted using a Dounce homogenizer. The membrane fraction was then collected by centrifugation at 100,000× g for 1 h and subsequently solubilized by addition of 1% (w/v) n-Dodecyl β-D-maltoside (DDM), 0.2% (w/v) cholesteryl hemisuccinate (CHS) for 2 h at 4 °C with rotation. The insoluble debris was removed by centrifugation at 100,000×g for 1 h. The supernatant was filtered and passed through Streptactin Beads (Smart-Lifesciences, China) pre-equilibrated with the buffer A (20 mM HEPES pH 7.5, 150 mM NaCl, 5 mM β-ME, 500 nM CaCl$_2$, and 0.01% (w/v) glyco-diosgenin (GDN)). Nonspecific bound protein was washed by using 10 column volumes of the purification buffer supplemented with 5 mM MgCl$_2$ and 5 mM ATP. The NHE1-CHP1 complex was eluted with the purification buffer supplemented with 5 mM desthiobiotin. The eluted protein sample was concentrated and further purified by gel filtration (Superose 6 Increase 10/300 GL, GE Healthcare, USA) pre-equilibrated in the corresponding purification buffer. Peak fractions were pooled and concentrated using a 100-kDa cut-off concentrator (Merck Millipore, Germany) to 3 mg/mL for nanodisc reconstitution.

For purification of cariporide-bound inhibited NHE1-CHP1 complex, 10 μM cariporide was supplemented to the culture during protein expression. Cells were collected and resuspended in a purification buffer B containing 20 mM HEPES pH 7.5, 100 mM KCl, 5 mM β-ME, 500 nM CaCl$_2$, and protease inhibitor cocktail, and the target protein complex was subsequently solubilized using the buffer B supplemented with 1% (w/v) DDM and 0.2% (w/v) CHS. Cariporide (10 μM final) was added throughout the purification; otherwise, the above described method was followed for the protein purification.

**Nanodiscs reconstitution.** The NHE1-CHP1 complexes together with POPC lipid were reconstituted into MSP1D1 nanodiscs at a final molar ratio (dimer-NHE1-CHP1: MSP1D1: POPC) of 1:5:50. Specifically, 25 mg/mL POPC was first suspended in the corresponding purification buffer supplemented with 2% (w/v) GDN. The mixture was incubated at 4 °C for 30 min to dissolve the suspended lipids. Purified NHE1-CHP1 complex (prepared at 1 nM) and MSP1D1 were added to the lipid in proportion and then gently tumbled at 4 °C for 1 h. To remove detergent, Biobeads SM2 were then added to the mixture (in a ratio of 400 mg beads per milliliter of mixture) and incubated at 4 °C for 2 h; repeated three times. After the final addition of Biobeads, the mixture was incubated overnight, tumbling end over end at 4 °C. The Biobeads were removed before the sfGFP-StrepII tag of NHE1 and the FLAG-8x His tag of CHP1 were digested with PPase. The digested protein sample was concentrated and loaded into a Superose 6 Increase column pre-equilibrated in the purification buffer without detergent. The peak fractions were pooled and concentrated to 5 mg/mL for cryo-EM sample preparation.

**Cryo-EM sample preparation and data acquisition.** Holey carbon grids (Au R1.2/1.3 300 mesh; Quantifoil) were glow-discharged in H$_2$-O$_2$ condition for 60 s using the Solarus plasma cleaner (Gatan, USA). A droplet of 2.5 μL of 6 mg/mL NHE1-CHP1 complex in nanodisc was applied on the glow-discharged grid. The grid was then blotted for 3.5 s at 4 °C under condition of 100% humidity using Vitrobot Mark IV (Thermo Fisher Scientific, USA), and flash-frozen in vitreous liquid ethane. Cryo-EM data were collected on a 300-kV Titan Krios or a 200-kV Talos Arctica (Thermo Fisher Scientific) using a K2 Summit direct electron detector positioned after a GIF quantum energy filter (Gatan). The slit of energy filter was set to 20 eV. Movie stacks were automatically collected using SerialEM[57] in super-resolution counting mode at magnification of 130,000×, yielding a calibrated super-resolution pixel size of 0.50−0.52 Å, with a nominal defocus range of 1.2−2.2 μm. Each movie stack was dose-fractionated in 32−42 frames with a total dose of 50−60 e$^-$/Å$^2$. The dose rate was 8.5−9.0 e$^-$/Å$^2$/s.

**Data processing.** All of the micrographs were motion-corrected and dose-weighted using MotionCor2[58] with 5 × 5 patching. The contrast transfer function (CTF) parameters for each micrograph were determined by GCTF[59]. Particles were picked using Gautomatch, Template Picker (cryoSPARC)[60] or Topaz[61]. All procedures of data processing were conducted in RELION-3.1[62] unless otherwise specified. For the NHE1-CHP1$^{Na/7.5}$ complex, a total of 1464k particles were picked from 2462 motion-corrected micrographs, followed by rounds of 2D classification to clean particles. Ab initio reconstruction was conducted in cryoSPARC to generate an initial map. Initial 3D classification against this map with C2 symmetry imposed without applying a mask generated eight classes. The most populated class was composed of 32% of total particles, featuring clearly resolved transmembrane helices and two CHP1 bound at intracellular side of the nanodisc.

Particles from this class were selected and submitted to further 3D auto refinement, yielding a 4.2-Å map. Another round of 3D classification using a mask which envelops the NHE1 TMD without alignment yielded a subset of 108k particles. The map was further improved by Bayesian polish and local CTF refinement, with a reported resolution of 3.3 Å according to the gold-standard *Fourier* shell correlation (GSFSC) criterion at the threshold of 0.143.

For the NHE1-CHP1$^{Na/6.5}$ complex, a total of 1,273k particles were initially picked from 2,205 micrographs, followed by particle cleaning using 2D classification in cryoSPARC. 3D classification against a low-resolution map of NHE1-CHP1$^{Na/7.5}$ complex was carried out in RELION with the application of C2 symmetry, which yielded eight classes. The most populated class displayed clearly resolved TM helices and was composed of ~41% of total particles. Particles from this class were selected and submitted to further 3D classification focusing on TMD domain. Four out of six classes displaying resolved transmembrane helices were selected for 3D refinement, yielding a 4.3-Å map. Another round of 3D classification focusing on the TMD without particle alignment was carried out, giving rise to a subset of 101k particles. Subsequent 3D refinement, in combination with Bayesian polish and CTF refine, yielded a 3.4-Å map according to the 0.143 cut-off of the GSFSC criterion[63]. To improve CHP1 density, 2-fold symmetry expansion was carried out using the particles from the 4.3-Å map of the overall structure. CHP1-focused 3D classification without the imposition of symmetry was then performed using a soft mask containing a single CHP at one side. One out of four classes showed obvious features of secondary structural elements, accounting for 24% of total input particles, and was subjected to further 3D refinement focusing on the NHE1 dimer and a CHP1. The resulting map has a reported resolution of 4.0 Å according to the GSFSC criterion. We also carried out similar analysis for the NHE1-CHP1$^{Na/7.5}$ complex, but we did not obtain a map with clear CHP1 density.

For the NHE1-CHP1$^{K/cariporide}$ complex, a total of 1314k particles were initially picked from 3855 motion-corrected micrographs, followed by 3D classification imposed with C2 symmetry using low-resolution NHE1-CHP1$^{Na/6.5}$ complex as the reference. 3D classification yielded eight classes. Class 5 (12%) and class 8 (11%) displayed clearly resolved TM helices. The densities of CHP1 resembled NHE1-CHP1$^{Na/6.5}$ complex in class 5. Particles from this class were selected for 3D refinement, resulting in a 4.3-Å map. Class 8 displayed a discernable different shape compared to the NHE1-CHP1$^{Na/6.5}$ complex, featuring the CHP1 movement towards the membrane plane. The obvious CHP1 secondary structural features in this initial 3D class suggest less conformation heterogeneity. This subset of particles was subjected to further 3D classifications, followed by Bayesian polish, CTF refine and auto refinement using a soft mask excluding nanodiscs, yielding a 3.9-Å map. To further improve the map quality of the NHE1-CHP1$^{K/cariporide}$ complex, particles were then imported to cryoSPARC and subjected to non-uniform refinement. The final map was reported at 3.5 Å resolution according to the GSFSC criterion.

**Model building**. De novo model building of TMD domain was initiated using the 3.4-Å map of the NHE1-CHP1$^{Na/6.5}$ complex in *Coot*[64]. Poly-alanine helices were manually placed and connected based on the map. The register assignment was based on the features of large aromatic sidechains. The manually built model was subjected to real space refinement in PHENIX against the final map of NHE1-CHP1$^{Na/6.5}$, with secondary structure, 2-fold non-crystallographic symmetry and Ramachandran restraints applied throughout the refinement[65]. To generate overall structure of the NHE1-CHP1 complex, the refined NHE1 model and NMR structure of CHP1-NHE1$^{514-545}$ (PDB ID: 2E30) were docked into EM map composed of a CHP1 and an NHE1 dimer, as rigid bodies. The cytoplasmic helix HC2 was built based on prominent α-helix density feature, immediately after HC1 and parallel to the membrane. The resulting model was refined against the NHE1-CHP1 complex map containing one NHE1 dimer and one CHP1. To generate a dimerized NHE1-CHP1 model (two NHE1 and two CHP1), we placed one more CHP1 molecular based on the 2-fold symmetry.

Model building of the NHE1-CHP1$^{Na/7.5}$ complex and NHE1-CHP1$^{K/cariporide}$ complex were started by docking NHE1-CHP1$^{Na/6.5}$ complex structure into corresponding maps. Manually adjustments were carried out aided by clear mainchain and sidechain features. In the NHE1-CHP1$^{K/cariporide}$ complex, the CHP1 showed distinct binding fashion to the NHE1 compared with NHE1-CHP1$^{Na/6.5}$ complex. One more cytoplasmic helix HC3 was placed right after HC2, supported by density feature and secondary structure prediction. The cariporide molecule was fitted into the map. The resulting models were subsequently refined against corresponding EM maps.

Figures were prepared using UCSF Chimera[66], UCSF ChimeraX[67], and PyMOL[68].

**Cell culture and western blotting**. Chinese hamster ovary AP-1 cells are a mutagenized cell line devoid of NHE1 expression[56]. AP-1 cells were cultured in α-MEM supplemented with 10% fetal bovine serum, penicillin (100 units/mL), streptomycin (100 μg/mL), and 25 mM NaHCO$_3$ (pH 7.4).

For western blot analyses, AP-1 were grown in 35-mm dishes and transiently transfected with 1.5 μg of plasmid DNA encoding wild-type (WT) NHE1 tagged at its C-terminus with mCherry (NHE1) alone or WT and mutant NHE1 constructs co-expressed with WT or mutant CHP1 inserted together in a pBIG1a vector-based expression cassette using Lipofectamine2000™ (Invitrogen) according to the manufacturer's recommended procedure. Cell lysates were prepared following 48 h post-transfection by washing cells twice on ice with ice-cold PBS, followed by scraping in 0.5 mL of lysis buffer (0.5% NP40/0.25% sodium deoxycholate/PBS supplemented with protease inhibitor cocktail (Roche Diagnostics). Lysates were incubated for 30 min on a rocker at 4 °C, and then centrifuged for 20 min at 4 °C to pellet the nuclei and cellular debris. In all, 20 μg of protein from the resulting supernatants were eluted in sodium dodecyl sulfate (SDS)-sample buffer (50 mM Tris-HCl, pH 6.8, 1% SDS, 50 mM dithiothreitol, 10% glycerol, 1% bromophenol blue), and subjected to 8% SDS-polyacrylamide gel electrophoresis (SDS-PAGE), then transferred to polyvinylidene fluoride (PVDF) membranes (Millipore, Nepean, Ontario, Canada) for immunoblotting. The membranes were blocked with 5% non-fat skim milk for 1 h, then incubated with the specified primary antibodies (mouse monoclonal NHE1 1:3000 (EMD Millipore Corp) and β-tubulin 1:10,000 (Sigma)) in PBS containing 0.1% Tween 20, followed by extensive washes and incubation with goat anti-mouse horseradish peroxidase (HRP)-conjugated secondary antibody (NHE1, 1:4000; β-tubulin, 1:10,000) for 1 h. Immunoreactive bands were detected using Western Lightning™ Plus-ECL blotting detection reagents (Perkin Elmer Inc., Waltham, MA).

**Immunofluorescence confocal microscopy**. AP-1 cells were cultured on fibronectin-coated 18-mm glass coverslips, transfected with the NHE1$_{ChFP}$ constructs, and fixed 48-h post-transfection with 4% paraformaldehyde for 20 min at room temperature. Nuclei were labeled with DNA dye DAPI (1:5000) for 15 min at room temperature. Coverslips were mounted and cells were examined by laser scanning confocal microscopy using the ZEN software of a Zeiss LSM 780 microscope equipped with a PMT detector, with images acquired using a ×63/1.4 NA oil immersion objective lens.

**Measurements of cytoplasmic pH**. AP-1 cells were grown in 35-mm dishes and transiently transfected with 2 μg of mCherry (ChFP), NHE1$_{ChFP}$ wild-type (WT) or the indicated mutants using Lipofectamine2000 according to the manufacturer's instructions. Five hours after transfection, cells were split and seeded onto fibronectin-coated (2 μg/mL in PBS pH 7.4 overnight at 4 °C) FluoroDishes™ and grown overnight.

To measure cytoplasmic pH, the cells were washed in Na$^+$-rich solution (140 mM NaCl, 5 mM KCl, 1 mM CaCl$_2$, 1 mM MgCl$_2$, 10 mM glucose, and 10 mM HEPES-Na$^+$, pH 7.4) three times and incubated with the pH-sensitive dye BCECF-AM (2′7′-bis-(2-carboxyethyl)−5-(and 6)-carboxyfluorescein-acetoxymethyl ester) (5 μM) diluted in Na + -rich solution for 30 min at 37 °C. Cells were subsequently washed four times in Na + -rich solution and subjected to ratiometric imaging using an inverted fluorescence microscope (IX81, Olympus, Center Valley PA) and imaging system (Photon Technology International, Edison NJ). Experiments were carried out at 37 °C using a thermostatically controlled platform (FC-5, Live Cell Instr., Seoul Korea). Flow rate was maintained at 1 mL/min using a four-channel peristaltic pump (205 S, Watson Marlow, Wilmington MA). Intracellular BCECF was excited alternately at 440 and 490 nm every 10 s. Transfected cells were identified by detecting mCherry fluorescence. The fluorescence intensities were measured using Easy Ratio Pro software (Horiba Sci. Mississauga, ON). Fluorescence ratios (F490/F440) were recorded and displayed continuously. The F490/F440 ratios were converted to pH using the K$^+$-nigericin clamp method[69]. Briefly, AP-1 cells were loaded with BCECF-AM and in situ calibration was performed by clamping the cytoplasmic pH between 6 and 8.5 in K$^+$-rich solution (135 mM KCl, 10 mM NaCl, 20 mM HEPES, 1 mM MgCl$_2$, and 0.1 mM CaCl$_2$) containing 10 μM nigericin and recording the fluorescence ratios.

To measure NHE1 activity, AP-1 transfectants were subjected to an NH$_4$Cl-induced acid-load[70,71] and then the initial rates of Na$^+$-dependent pH recovery were measured as a function of time (ΔpH/min). The rates were calculated based on the initial linear phase of the pH recovery. Briefly, baseline fluorescence ratios were recorded for 5 min in Na$^+$ rich solution, followed by an acute acid load induced by incubating the cells for 5 min in isotonic NH$_4$Cl solution (50 mM NH$_4$Cl, 70 mM choline chloride, 5 mM KCl, 1 mM MgCl$_2$, 2 mM CaCl$_2$, 5 mM glucose, 20 mM HEPES-Tris, pH 7.4), followed by 10 min in isotonic Na$^+$-free solution (125 mM choline chloride, 1 mM MgCl$_2$, 2 mM CaCl$_2$, 5 mM glucose, and 20 mM HEPES-Tris, pH 7.4). pH recovery was measured for 15 min upon reintroduction of Na$^+$-rich solution. Graphics were performed using OriginPro 2020 software.

**Reporting summary**. Further information on research design is available in the Nature Research Reporting Summary linked to this article.

## Data availability

The three-dimensional cryo-EM density maps of the NHE1-CHP1$^{Na/7.5}$ complex, NHE1-CHP1$^{Na/6.5}$ complex and NHE1-CHP1$^{K/cariporide}$ complex have been deposited in the EM Database under the accession codes EMD-30848, EMD-30847, and EMD-30849, respectively, and the coordinates for the structures have been deposited in Protein Data Bank under accession codes 7DSW, 7DSV, and 7DSX, respectively.

Source data are provided with this paper.

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

## Acknowledgements

We thank X. Huang, B. Zhu, X. Li, L. Chen, and other staff members at the Center for Biological Imaging (CBI), Core Facilities for Protein Science at the Institute of Biophysics, Chinese Academy of Science (IBP, CAS) for the support in cryo-EM data collection. We thank Yan Wu for his research assistant service. This work is funded by Chinese Academy of Sciences Strategic Priority Research Program (Grant XDB37030304 to Y.Z., XDB08020301 and XDB37030301 to X.C.Z.) and National Natural Science Foundation of China (Grant 31971134 to X.C.Z.), We are grateful for the services provided by McGill Life Sciences Advanced BioImaging Facility and Genome Québec for DNA sequencing; platforms supported by funding from the Canadian Foundation for Innovation. We also kindly thank J. Hanrahan (Department of Physiology, McGill University) for providing access to the pH imaging system and funding from the Canadian Institutes of Health Research (Grants PJT-155976 and PJT-166165).

## Author contributions

Y.Z. conceived the project. Y.D. prepared sample for cryo-EM study and made all of the mutation constructs. Y.D., Y.G., and Y.Z. collected cryo-EM data. Y.G. and Y.Z. calculated the EM maps. Y.G., B.L., and Y.Z. built and refined the atomic model. A.I., D.K., A. B., and J.O. performed AP-1 cell-based NHE1 expression, confocal microscope imaging, and transport assay. X.Z. and Y.Z. analyzed the data. All authors contributed to manuscript preparation.

## Competing interests

The authors declare no competing interests.
