## [Peer Review File · Nature Communications]

REVIEWER COMMENTS

Reviewer #1 (Remarks to the Author):

Reviewer comments to Dong et al. Structure and mechanism of the 1 human NHE1-CHP1 complex

General

This well-written manuscript reports the single-particle cryo-EM structure of human SLC9A1 (NHE1) in complex with CHP1, in the inward as well as in the outward facing, inhibitor-bound conformation. The study confirms and extends evidence from biochemical and cell biological work showing that NHE1 is a dimer, and shows that during transport, the structure undergoes the same type of elevator-like movement demonstrated previously for bacterial homologs.

The approach of co-expressing and reconstituting the NHE1-CHP1 complex in nanodiscs and using cryo-EM to obtain the structures +/- inhibitor for stabilizing the outward open conformation is very sensible and has allowed the team to successfully solve this long-awaited structure. The experiments are generally well carried out and the results are important and should make a substantial impact in the field, as the first structure of a mammalian NHE1. That said, I do have some concerns and suggestions for the authors to consider, listed below.

Major

1. To ensure that the structure represents the correctly folded protein, it is essential to show that the reconstituted NHE1 possesses transport activity, for instance in a liposome reconstitution assay.
2. Both proteins were tagged for purification purposes, but it seems that only the GFP tag on NHE1 was removed, while the 8xHis and FLAG tags on CHP1 were not? If this is correctly understood, please comment on how this might have affected the structure. If not, please specify in the materials and methods section.
3. Interestingly, in the NHE1 dimer structure presented, H2 and H3 in the membrane-proximal part of the tail are folded under NHE1 in some states, H3 interacting with CHP1 in a manner proposed to be critical for NHE1 ion exchange. In previous works, these helices were shown to interact with the inner membrane face. This is enticing, but also raises several technical questions: Firstly, could the dimensions of the nanodisc have affected the structure, i.e. that if there was insufficient free lipid surface for this part of the tail to interact with, it might erroneously have interacted with the CHP? The authors should provide a calculation of whether the disc is sufficiently large to accommodate the dimer, the CHPs, and the association of the tails with the remaining space. Second, NHE1 is highly dependent on interactions with PIP2 in the plasma membrane, yet the lipids in the nanodisc were pure POPC. I realize that asymmetric nanodiscs would be a tall order, but how sure are the authors that the lack of more realistic inner leaflet lipids did not affect the structure, in particular the CHP1-NHE1 tail complex? This should be commented on in the revised manuscript.
4. Perhaps I missed something, but if the interaction between H3 and CHP1 is critical for ion exchange, why does the R192A CHP1 mutation have no effect? As for the activity of the Y577A/H578 mutation decreasing transport "by 50% relative to the WT complex" (p 17, l 480) inspection of figure S8C indicates to me that the expression level of the mutant is about 50% of that of the WT (although difficult to see because the WT band is saturated, which would only make the difference greater). So how sure are the authors that this is critical for function? Please quantify the expression data.
5. It is interesting that the structure is unresolved N-terminally to EL1 (given the extensive literature suggesting the existence of a TM1 prior to this – which would not offhand be expected to exhibit so high flexibility that it would not be resolved. How do the authors interpret this? Please provide a brief discussion.

6. The organization of the TM helices in the present structure differs profoundly from the classical view of NHE1 from the Wakabayashi group, which was based on substantial experimental evidence of residue accessibility (J Biol Chem 275: 7942–7949, 2000). I do not at all argue that the present structure may be the correct one – but a discussion of how the accessibility results from previous work can be reconciled with the present structure is warranted.

7. The purified NHE1 obtained is a mix of the core- and fully glycosylated NHE1 forms. How might this have affected the structure, and – given that the glycosylations are found in EL1, part of which is not present in the model – do you have any indication of whether the glycosylation is important for the NHE1- and specifically EL1 - structure?

Minor

P 2, l 49-54. The way the sentence is structured, it provides the first suggestion of which I am aware, that *Equus caballus* is a bacterial species :-).

P 7, l 165. -> well resolved

P 15, l 415 ff. I may have missed it but I did not find a reference cited for the critical function of E262 – this should be provided.

Reviewer #2 (Remarks to the Author):

Sodium ion/proton exchangers are highly important membrane integrated transporters controlling the cytoplasmic pH and the cell volume. The electroneutral exchangers from eukaryotes are called NHE 1 - 13. NHE1 is also the target of important drugs. Scientifically the mechanism of exchange is highly debated. There is only one structure available, that of NHE9 from horses in one conformation. The authors present the structure of NHE1 from humans, and that in two different conformations, one cytoplasmically open, and one externally open in the presence of an inhibitor. The comparison of both comparison adds strong evidence for an elevator-type of mechanism where part of the protein undergoes an elevatorlike movement enabling access of the substrate binding site to the from the other side of the membrane. NHE1 has been studied by many groups, the key of the success of the authors obviously was the coexpression of NHE1 with a CHP as a functionally important binding partner. The paper is written in a very clear way and absolutely convincing. The structure was determined appropriately. The accurate description of the methods used should allow to reproduce the results. The discussion is convincing. I do not have to criticize much. My major concern is that the figures are too small and barely readable even with my strongest glasses. I suggest to increase there size by 50 to 100 %. There also could be a more detailed comparison with NHE9 (not necessarily)

minor points:

line 243 replace "involved" by "involving"

line 651 51 ? please clarify

Reviewer #3 (Remarks to the Author):

The structure of NHE1 in complex with CHP1, an inhibitor and in several conformational states is a fantastic achievement and will be well-appreciated in the community. The cryo EM structural work appears mostly solid and the functional data complements some of the more novel findings. However, the current manuscript is somewhat let down by figures where it is hard to interpret the structural

details and conclusions that are difficult to follow in places.

Major concerns:

1. The authors have reached their mechanistic conclusions by comparing the outward and inward facing NHE1-CHP complex structures. While understandable, the problem is that the authors appeared to have made the assumption that the inhibitor bound outward-facing NHE1-CHP structure is overall equivalent to NHE1-CHP structure without inhibitor bound. However, since the inhibitor makes interactions between both the dimer and mobile core domains, its likely that the inhibitor has captured a NHE1 state wherein the CTD domain interact differently than how they would during an NHE transport cycle. I think its more than likely that in the inhibitor-bound stabilised NHE1 state the CTD region is less mobile and has enabled HC3 to interact with HC3 from the other promoter. While I agree that the inhibitor bound structure has probably enabled a glimpse at a allosteric mechanism and its important, the mechanistic interpretations one can derive from this need to be better clarified. Indeed the CTD is not required for exchange activity.

2. Linked to the question above, what is the proposed model for CHP1 activation based on these two structures? In the inhibitor bound state the C-terminal HC3 helices come together. It wasn't clear to me whether in this conformation there was still space for the core transport domains to transition into an inward-facing state or not? Does CHP1 remove a potential auto-inhibitory state and therefore increases NHE1 activity or does it promote NHE1 activation by stabilising the homodimer? It is essential to clarify the proposed model in the text. Furthermore, mutations disrupting the HC3 interactions between the two promoters would help to tease apart mechanistic details, i.e., does activity increase or decrease when HC3 can no longer interact with HC3 in the neighbouring promoter?

3. I suspect that this paper was likely written up before the cryo EM structure of NHE9. As such, the paper has more comparisons between NHE1 and the bacterial homologues than it does between NHE1 and NHE9. This is a shame as NHE1 is the first structure of a plasma membrane localised NHE, which is evolutionary quite divergent from NHE9, which is predominantly localised to organelles. An obvious question is what are the structural differences causing the large sequence divergence between these two NHE isoforms? For example, the model of NHE1 based on NHE9 indicated that the intracellular cavity was much more negatively charged than NHE9 (EMBO J (2020)39:4541-4559). Furthermore, the NHE9 structure has already established the conserved elevator mechanism with the bacterial Na⁺/H⁺ antiporters and while the NHE1 structures help to refine this model, it nevertheless takes the focus away from the more interesting questions regarding allosteric regulation and inhibition.

4. For cryo EM processing there are several stages that I found difficult to follow. In ED.Fig.2, they chosen one class from 3D classification and refined, but the CHP1 density has disappeared. Was this density lost due to averaging or masking? Again in ED.Fig.3 the CHP1 density is a bit odd. Before focused classification for CHP1 there is no density for CHP1, but after focused classification the density comes back and looks like an artifact? Please clarify more and make it clear at each step where a mask has been used or not.

Minor concerns.

1. I do not think that the elevator mechanism is "controversial" for electroneutral transporters vs. electrogenic transporters. The elevator mechanism was first proposed by the outward facing state of NapA (Nature (2013)) and confirmed by NapA states in opposite facing conformations (NSMB (2016)). Given the high degree of structural conservation between bacterial electroneutral and electrogenic transporters it is expected all would work with the same type of alternating-access mechanism as recently confirmed (EMBO J (2020)39:4541-4559). Rather than "controversial" I would say the the elevator alternating access mechanism in the NHEs needs to be better refined.

To be clear the only other alternative to an elevator mechanism would be a rocking-bundle model. The difference between a rocking-bundle and elevator mechanism is not decided if the structural transitions "look" more like rocking or vertical movements. The distinction between the two models of

alternating access is that in elevator proteins the substrate binds to only ONE of the two domains and then this ONE domain carries the substrate some vertical distance across the membrane, i.e., in contrast to rocking bundles where the substrate binds between two domains and the domains move around the substrate (Annual Review of Biochemistry Vol. 85:543-572). The additional exchanger structures enable one to refine the elevator alternating-access mechanism. The recent cryo EM structure of NHE9 enables some refinement by comparing to bacterial homologues and clearly the NHE1 structures can add more insight, in particular, in regards to allosteric regulation via CHP1 binding to the CTD domain.

2. Is there any coupling between the inhibitor cariporide and CHP1? Is there still inhibition of NHE1 when the C-terminal tail is truncated? Does the inhibitor conformationally stabilise NHE1? In other words, is part of its ability to inhibit NHE1 caused by the fact that it restricts NHE1 mobility by also interacting with the N-terminus of the neighbouring protomer and also the dimer domain (E346 in TM8)? Does the N-terminal tail change its position when comparing the inhibitor and non-inhibitor bound states?

3. A superimposition of the outward and inward facing states results in a 5 Å vertical displacement of the strictly conserved ion-binding D267 based on its Calpha position. This displacement is consistent with the 5Å displacement modelled between inward facing NHE9 and an outward model based on NapA (EMBO J (2020)39:4541-4559). However, NHE9 is less inward facing than inward NapA and, as such, the vertical displacement between opposite facing NapA states is larger at 8Å between Calpha positions and closer to 10Å when the Cbeta positions are included (NSMB (2016)). Part of the reason between larger differences between Calpha and Cbeta positions in NapA is that the ion binding aspartate was found to rotate inwards between outward and inward facing conformations in NapA, which could be mechanistically important, i.e., it was proposed that neutralisation of the ion-binding aspartate upon Na⁺ binding would facilitate the transition across the hydrophobic barrier of the dimerization domain surface (NSMB (2016)). Is there any rotation observed of the ion-binding aspartate between opposite facing conformations and what is the Cbeta distance? I understand if this cannot be determined due to a lack of map density for the aspartates (negative scattering factors).

4. I was a bit confused by the sentence "However, since the conformational change during the state transition is a relative one between the core and dimerization domains, an alternative movement cannot be ruled out." Is this meant to say that the transport domains stay's fixed and the dimer domains move? Conceptually, this would not make any sense, since the substrate doesn't bind to the dimer domains, but only the core domains and in this way the substrate is carried across the membrane....

... there is no evidence that ion-binding stabilise the core domains, but the opposite (eLife. 2014; 3: e01412.). Furthermore, from MD simulations of NapA it was modelled the the dimer domain bobs up-and-down a bit, but the transport domains move up and down considerably much more (NSMB (2016)). Indeed intrinsic dynamics of NHE9 show that the core domains are very mobile and spontaneous cover 80% of the transitions observed between NapA crystal structures in opposite facing states (EMBO J (2020)39:4541-4559). Also in the evolutionary related citrate transporter structure CitS, there is an asymmetric crystal structures available with one transport domain in the outward conformation and the other in the inward-facing conformation (eLife 2015;4:e09375). In these crystal structures the dimerization domain stays fixed. Taken together, and added to the fact that in the many other elevator proteins the oligomerization interface stays fixed, I think there is no evidence to suggest otherwise.

5. The salt-bridge between E262 and R425 is conserved in all electroneutral transporters (Nature Communications : 4205 (2018), as modelled and as shown in PaNhaP and NHE9 structures (EMBO J (2020)39:4541-4559). As such, E262 is not thought to be involved directly in ion-binding, but it could still be allosterically coupled to ion-binding. I wouldn't say that the integrity of the E262-R425 salt-

bridge is essential for the integrity of the ion-binding domain. In NapA outward and inward facing structures a parallel salt-bridge remains intact between opposite facing states (D156-K305) (NSMB (2016). However, this does not mean that this salt-bridge is "essential" to the core domain integrity as you can make mutations to the lysine and still achieve transport (PNAS February 14, 2017 114 (7) E1101-E1110), its just that most mutations are electroneutral unless the lysine is mutated to histidine. MD simulations in NhaA show that this corresponding salt-bridge is broken upon sodium binding (J Gen Physiol (2014) 144 (6): 529–544.). Its plausible that the E262-R425 salt-bridge is also broken during transport and, at the very least, an intact salt-bridge as seen here does not rule out this possibility.

6. The proposed rearrangement of TM5b (lines 420-433) doesnt make sense to me and we have not seen this before in any of the previous inward-facing Na⁺/H⁺ antiporter structures. I would argue that the cryo EM density is probably not good enough to assign TM5b geometry properly due to its mobility and its been extended during real-space refinement. A similar issue for this half-helix was also apparent in low resolution NhaA crystal structures (J Gen Physiol (2014) 144 (6): 529–544).

7. Was it really the complete full-length NHE1 expression? SDS-page is not sensitive enough to reach this conclusion. Based on the cryo EM structure of NHE9 and sequence analysis it was proposed that NHE1 likely has a cleavable signal peptide rather (EMBO J (2020)39:4541-4559), rather than 14 TMs. Interestingly in the NHE1 structure, residues 1 to 87 aa could not be modelled, which would be the most consistent with this analysis. Do you think NHE1 has a cleaved signal peptide and have you carried our native MS to ascertain otherwise?

8. Related to previous question is whether or not the N-terminal tail reaching across to the other promoter (residues 87 to 99) is a form of allosteric coupling between the promoters? As a side note, I would say its unclear if this region and possible regulation is present in NHE9 or not as the first 20 or so amino acids could not be modelled (EMBO J (2020)39:4541-4559). I think the term "structural swapping" is a bit confusing, since its not swapped protein domain, but rather an extended polypeptide that reaches across to the other promoter.

9. The superimposition is calculated for NHE1 vs NHE9 dimers. What are the RMSD differences between NHE9 vs NHE1 monomers? Does the structural divergence come from differences in oligomerization or local differences within the promoter that could be informative?

10. In NHE9 it was proposed that PIP2 lipids bind at the hydrophobic interface to stabilise the homodimer. However, the PIP2 binding seemed to require an loop domain, which is not present in NHE1. Is there any clues as to what lipids bind at the interface?

11. Related to the above question PIP2 is reported to be bind at IL7. I found it difficult to follow the authors arguments of how PIP2 binding could increase NHE1 activity.

12. Please include the following reference first proposing an elevator mechanism for Na⁺/H⁺ exchangers and indeed the first evidence that an elevator model is a general type of alternating-access mechanism for secondary-active transport; as first observed in the glutamate transporter GltPh.

"A two-domain elevator mechanism for sodium/proton antiport". Nature. 2013 Sep 26; 501(7468): 573–577.

13. I noticed that the first low resolution cryo EM electron crystallography structure of NhaA was referenced instead of the NhaA crystal structure. I think the later is a more appropriate reference as it enabled the first glimpse into the overall NhaA-fold as shared by all Na⁺/H⁺ exchangers (Nature 435 (2005), 1197–1202) and even Na⁺-coupled bile acid symporters (Nature. 2011 Oct 5; 478(7369): 408–411).

To Reviewer #1

Major

Comment 1) To ensure that the structure represents the correctly folded protein, it is essential to show that the reconstituted NHE1 possesses transport activity, for instance in a liposome reconstitution assay.

Reply: We agree with the reviewer in that it would be nice to study function of the purified protein sample by in vitro transport assays. However, it proved to be highly challenging due to the low yield of the recombinant NHE1-CHP1 complex, whereas liposome reconstitution would require large amounts of the protein sample. On the other hand, our structures are consistent with abundant previous studies. Both the overall structure and critical residues involved in cation binding in our structure are superimposable to those from other homologs. More importantly, our structures support previous functional analyses on various NHE1 mutations, including residues involving ion exchange and inhibitor binding. Taken all together, we are confident that our purified sample used for the cryo-EM study represents its native conformation(s).

Comment 2) Both proteins were tagged for purification purposes, but it seems that only the GFP tag on NHE1 was removed, while the 8xHis and FLAG tags on CHP1 were not? If this is correctly understood, please comment on how this might have affected the structure. If not, please specify in the materials and methods section.

Reply: We thank the reviewer(s) for pointing this out. In fact, we also included the PPase cleavage site before the His-FLAG tag in the CHP1 construct, and both of the GFP-StrepII double tag of the NHE1 and the 8xHis-FLAG double tag of the CHP1 construct were removed by incubation with PPase in our experiment. We clarify this in the revised Method section.

In the revised line 509, it now reads “The CHP1 gene was similarly cloned into a modified pEG BacMam vector, with a PPase cleavage site, 8xHis and FLAG tags being fused to the C-terminus of the gene product.”

In the revised line 557, it now reads “The Biobeads were removed before both the sfGFP-StrepII double tag of the NHE1 sample and the FLAG-8xHis double tag of the CHP1 construct were digested with PPase.”

Comment 3) Interestingly, in the NHE1 dimer structure presented, H2 and H3 in the membrane-proximal part of the tail are folded under NHE1 in some states, H3 interacting with CHP1 in a manner proposed to be critical for NHE1 ion exchange. In previous works, these helices were shown to interact with the inner membrane face. This is enticing, but also raises several technical questions: Firstly, could the dimensions of the nanodisc have affected the structure, i.e. that if there was insufficient free lipid surface for this part of the tail to interact with, it might erroneously have interacted with the CHP? The authors should provide a calculation of whether the disc is sufficiently large to accommodate the dimer, the CHPs, and the association of the tails with the remaining space. Second, NHE1 is highly dependent on interactions with PIP2 in the plasma membrane, yet the lipids in the nanodisc were pure POPC. I realize that asymmetric nanodiscs would be a tall order, but how sure are the authors that the lack of more realistic

inner leaflet lipids did not affect the structure, in particular the CHP1-NHE1 tail complex? This should be commented on in the revised manuscript.

Reply: We appreciate the reviewer's comment. The dimension of the nanodisc and NHE1 TMD are approx. $109 \text{ \AA} \times 129 \text{ \AA}$ and $50 \text{ \AA} \times 90 \text{ \AA}$, respectively, suggesting that the nanodisc maintains large margins of lipid bilayer surrounding the NHE1 dimer. As shown in Figure A below, there is sufficient lipid surface area for HC1/HC2/HC3 to interact with if they preferred to do so. However, only HC2 is possibly directly contact with the membrane, consistent with its amphipathic property. Moreover, the CHP1-HC3 interaction predominantly exists in the outward-facing conformational state, which allowed us to unambiguously resolve the structures of both CHP1 and HC3, and we believe that this interaction is physiologically relevant. Importantly, disruption of the CHP1-HC3 interaction decreases the activity in our in vivo assay, further confirming the significance of this interaction.

Figure A. Cryo-EM map of the NHE1-CHP1 complex at outward-facing conformational state viewed from extracellular side (left) and intracellular side (right). Nanodisc and NHE1-CHP1 are colored by grey and cyan, respectively. HC2 and HC3 are highlighted by orange and green, respectively.

Whereas it has been proposed that PIP₂ plays an important role for the optimal activity of NHE1, PIP₂ is not essential. Depletion of PIP₂ reduces the pH-sensitivity of the exchanger towards the acidic range, such that activation of NHE1 requires a high H⁺ concentration (Aharonovitz et al. JCB, 2000). We thus do not expect that PIP₂ binding would significantly alter the transporter conformation, although it may induce some local conformational changes to fine tune the activity. Three regions have been proposed to potentially associate with PIP₂, including residues on IL7, IL8 and HC2, which are all close to the CHP1 binding site. In the inward-facing conformation, the CHP1 appears to be highly mobile in the absence of PIP₂. Therefore, we hypothesize that the PIP₂ binding regulates the transport activity by affecting the CHP1 conformation. We have added a brief discussion in the revised manuscript. Please see Pg 17 (lines 470-476). It reads "... So far, residues on IL7, IL8, and HC2 have been identified to be critical for PIP₂ modulation^{54,55}, which are all close to the CHP1 binding site. Hence, we proposed that PIP₂ modulates the activity of NHE1 by affecting CHP1 conformation, particularly for the CHP1 in the inward-facing conformation which exhibits highly motility relative to NHE1 in the absence of PIP₂. The structure of a PIP₂-bound NHE1-

CHP1 complex is required to fully understand the modulation mechanism of PIP₂ on NHE1 transport activity.”

Comment 4) Perhaps I missed something, but if the interaction between H3 and CHP1 is critical for ion exchange, why does the R192A CHP1 mutation have no effect? As for the activity of the Y577A/H578 mutation decreasing transport “by 50% relative to the WT complex” (p 17, line 480) inspection of figure S8C indicates to me that the expression level of the mutant is about 50% of that of the WT (although difficult to see because the WT band is saturated, which would only make the difference greater). So how sure are the authors that this is critical for function? Please quantify the expression data.

Reply: We appreciate this comment. Based on structural analysis of the outward-facing NHE1-CHP1 complex, each CHP1 molecule interacts with two distinct regions of the NHE1, including TM11 and HC3. Residues Y577/H578^{NHE1} and Q151/I155^{CHP1} are critical for the CHP1-HC3 interaction. Disruption of this interaction resulted in decrease of the transport activity, indicating that the CHP1-HC3 interaction is important for the transport function. On the other hand, R192^{CHP1} does not participate in the CHP1-HC3 interaction. Instead, it is involved in a charge-dipole interaction between CHP1 and TM11. There is little effect of mutation R192A on the transport activity. The related description please find in manuscript line 457-460. It reads “...For instance, the sidechains of Q151^{CHP1} and I155^{CHP1} form a hydrogen bond and hydrophobic interaction with H578^{NHE1} and Y577^{NHE1}, respectively; and R192^{CHP1} forms a charge-dipole interaction with the C-cap of TM11 (Figs. 5d and 5e).”

Regarding concerns about levels of NHE1 expression, protein expression in 5 separate transfection experiments was determined by densitometry of X-ray films exposed within the linear range of the films. To control for subtle loading variations, NHE1 expression was determined relative to endogenous β -tubulin levels (ratio of NHE1/ β -tubulin) and the resulting ratio values were normalized to 100%. These new data are now included in the Extended Figure 8 as panel d. These data show that there is no statistical difference in expression levels between NHE1-CHP1 and Y577/H578^{NHE1}-CHP1, suggesting the mutation has indeed affected the exchanger’s transport activity. To further illustrate the changes in pH recovery of NHE1 and the various mutant constructs, rather than showing the entire NH₄Cl-imposed acid-load and pH recovery over a 35-40 min, we have presented just the pH-recovery phase (4 min). This is now shown in Extended Figure 8 as panel e.

5. It is interesting that the structure is unresolved N-terminally to EL1 (given the extensive literature suggesting the existence of a TM1 prior to this – which would not offhand be expected to exhibit so high flexibility that it would not be resolved. How do the authors interpret this? Please provide a brief discussion.

Reply: We appreciate the reviewer’s comment. It remains controversial about the existence of a TM helix in the unresolved N-terminal domain (UNTD). In the paper mentioned by the reviewer in Comment 6, Wakabayashi’s group proposed that there is a putative TM helix in the UNTD. However, the N terminal of NHE1 is predicted to be a cleavable signal peptide using SignalP-5.0, instead of a transmembrane helix (Figure B). Moreover, a homolog of NHE1, NHE3 was confirmed to have a cleaved N terminal peptide (Zizak, M., et al. *Biochemistry*, 39(27), 8102-8112.). To clearly address this question, we attempted to carry out an N-terminal sequencing using our cryo-EM sample. Unfortunately, the Edman degradation was failed because the NHE1 has a blocked N terminus. And it is not feasible to determine N terminal sequence using mass spectrometry because

of poor compatibility of mass spectrometry with detergents. We have added a brief discussion. Please also see our reply to comment 6.

Figure B. Signal peptide prediction result using SignalP-5.0 Server

Comment 6) The organization of the TM helices in the present structure differs profoundly from the classical view of NHE1 from the Wakabayashi group, which was based on substantial experimental evidence of residue accessibility (J Biol Chem 275: 7942–7949, 2000). I do not at all argue that the present structure may be the correct one – but a discussion of how the accessibility results from previous work can be reconciled with the present structure is warranted.

Reply: We agree with the reviewer in that it is important to compare our structures with a prevalent topology model determined by Wakabayashi group (Wakabayashi’s model). Basically, we think that the topology of the NHE1 achieved from the current cryo-EM study is mostly consistent with Wakabayashi’s model, except for two discrepancies. The first discrepancy is the hypothetical TM helix in the UNTD, which is more likely to function as a signal peptide. (Please also see reply to Comment 5.) The other minor discrepancy is the reentrant loop between TM9 and TM10 in Wakabayashi’s hypothetical model, which is actually folded into two short membrane spanning helices, denoted as TM10 and TM11 in our structures. Nevertheless, we consider that these two helices are compatible with the reentrant loop of Wakabayashi’s model. We have added a short discussion in our manuscript. Please see Pg 7 (lines 164-170). It reads “The topology of the NHE1 determined in current study is mostly consistent with a previous prevalent model assessed by substituted cystine accessibility analysis³⁵. One of major discrepancy is the existence of a TM helix in the unresolved N-terminal region, which is predicted to be a cleavable signal peptide using SignalP-5.0 Server³⁶. The other minor discrepancy is the reentrant loop between TM9 and TM10 in the Wakabayashi’s hypothetical model, which is actually arranged into two short TM helices (TM10-11) in our structure.”

Comment 7) The purified NHE1 obtained is a mix of the core- and fully glycosylated NHE1 forms. How might this have affected the structure, and – given that the glycosylations are found in EL1, part of which is not present in the model – do you have any indication of whether the glycosylation is important for the NHE1- and specifically EL1 - structure?

Reply: The NHE1 contains N-linked glycosylation at N75 in the UNTD region. However, this region is unresolvable in the current cryo-EM study, indicating high flexibility and being consistent with the secondary structure prediction that it is a non-structured region. Previous studies have shown that removal of the N75 glycosylation site by site-directed mutagenesis produced a functional exchanger that had transport rates and pharmacological profiles similar to wild-type (Counillon, L., et al. (1994). *Biochemistry* 33: 10463-10469). Therefore, we do not anticipate that the glycosylation in this region would profoundly change the NHE1 or its EL1 structure.

Minor

Comment 8) P 2, l 49-54. The way the sentence is structured, it provides the first suggestion of which I am aware, that *Equus caballus* is a bacterial species :-).

Reply: We adjusted our statement. It now reads "... and most recently mammalian NHE9 from *Equus caballus* (*EcaNHE9*)".

Comment 9) P 7, l 165. -> well resolved

Reply: We thank the reviewer for pointing out this and have made a correction.

Comment 10) P 15, l 415 ff. I may have missed it but I did not find a reference cited for the critical function of E262 – this should be provided.

Reply: We appreciate the reviewer's comment and have added a reference.

To Reviewer #2

Sodium ion/proton exchangers are highly important membrane integrated transporters controlling the cytoplasmic pH and the cell volume. The electroneutral exchangers from eukaryotes are called NHE 1 - 13. NHE1 is also the target of important drugs. Scientifically the mechanism of exchange is highly debated. There is only one structure available, that of NHE9 from horses in one conformation. The authors present the structure of NHE1 from humans, and that in two different conformations, one cytoplasmically open, and one externally open in the presence of an inhibitor. The comparison of both comparison adds strong evidence for an elevator-type of mechanism where part of the protein undergoes an elevatorlike movement enabling access of the substrate binding site to the from the other side of the membrane. NHE1 has been studied by many groups, the key of the success of the authors obviously was the coexpression of NHE1 with a CHP as a functionally important binding partner.

The paper is written in a very clear way and absolutely convincing. The structure was determined appropriately. The accurate description of the methods used should allow to reproduce the results. The discussion is convincing. I do not have to criticize much. My major concern is that the figures are too small and barely readable even with my strongest glasses. I suggest to increase their size by 50 to 100 %. There also could be a more detailed comparison with NHE9 (not necessarily).

Reply: We appreciate very much the reviewer's positive comments.

Minor points

Comment 1) line 243 replace "involved" by "involving"

Reply: Change has been made.

Comment 2) line 651 51 ? please clarify

Reply: It (#51) was a reference number. We have reformatted it.

To Reviewer #3

Major concerns

Comment 1) The authors have reached their mechanistic conclusions by comparing the outward and inward facing NHE1-CHP complex structures. While understandable, the problem is that the authors appeared to have made the assumption that the inhibitor bound outward-facing NHE1-CHP structure is overall equivalent to NHE1-CHP structure without inhibitor bound. However, since the inhibitor makes interactions between both the dimer and mobile core domains, its likely that the inhibitor has captured a NHE1 state wherein the CTD domain interact differently than how they would during an NHE transport cycle. I think its more than likely that in the inhibitor-bound stabilized NHE1 state the CTD region is less mobile and has enabled HC3 to interact with HC3 from the other promoter. While I agree that the inhibitor bound structure has probably enabled a glimpse at a allosteric mechanism and its important, the mechanistic interpretations one can derive from this need to be better clarified. Indeed the CTD is not required for exchange activity.

Reply: We appreciate the reviewer's comment. The inhibitor cariporide does bind at the interface between the dimerization domain and core domain. However, while some side chains of critical residues are obviously altered, the main chains of the dimerization and core domains are not changed very much upon inhibitor binding, evidenced by the R.M.S.D. of the dimerization domain and of the core domain between the two distinct conformational states. They are 1.78 Å and 1.24 Å for the NHE1 monomer, respectively, and 1.92 Å and 3.41 Å for the NHE1 dimer, respectively, indicating that the inhibitor binding induces the core domain conformational change as a rigid body movement. Thus, it is reasonable to propose the elevator like transport mechanism based on comparing inhibitor bound and unbound states using the dimerization domain as the structural reference.

In the cariporide bound outward-facing conformational state, the dimerized HC3 helices are fixed between NHE1 and CHP1. Nevertheless, we do not propose that the inhibitor cariporide directly exerts effect on the CTD and decreases its motility, as the inhibitor and CTD are located at opposing sides of the membrane. Interestingly, a previous study (Hisamitsu, T et al., *Biochemistry* 2004, 43(34), 11135–11143) has demonstrated that recombinantly expressed isolated CTD is prone to form a dimer and the HC3 region is critical for the CTD dimerization. Thus, we suspect that dimerization of the HC3 is independent on the conformational state and inhibitor binding.

The CTD is responsible for the binding of a variety of regulatory proteins. Although the reviewer suggests that it may not be essential, we prefer to think that CTD is important for regulating the

transport activity. At least, deletion of the HC3 region decreases transport activity (Hisamitsu, T et al., *Biochemistry* 2004, 43(34), 11135–11143).

Comment 2) Linked to the question above, what is the proposed model for CHP1 activation based on these two structures? In the inhibitor bound state the C-terminal HC3 helices come together. It wasn't clear to me whether in this conformation there was still space for the core transport domains to transition into an inward-facing state or not? Does CHP1 remove a potential auto-inhibitory state and therefore increases NHE1 activity or does it promote NHE1 activation by stabilising the homodimer? Is it essential to clarify the proposed model in the text. Furthermore, mutations disrupting the HC3 interactions between the two promoters would help to tease apart mechanistic details, i.e., does activity increase or decrease when HC3 can no longer interact with HC3 in the neighbouring promoter?

Reply: CHP1 is observed to differentially associate with NHE1 in the two conformational states. In the inward-facing conformation, the CHP1 molecule exhibits high motility relative to NHE1. The motility of CHP1 is essential in this case either for its pH sensing ability or due to adjusting critical regulatory factors, such as PIP2 or other protein partners. In sharp contrast, the NHE1-CHP1 forms a stable complex in the outward-facing conformation. We identified CHP1-HC3 interaction, which is likely to be critical for the activity. Because it is exclusively found in the outward-facing conformation, we hypothesize that this interaction is important for CHP1 regulation of the transport activity by either facilitating the in-to-out conformational change or stabilizing the outward-facing conformation. To fully understand the CHP1 regulation roles, complex structures with PIP2 and/or other protein partner(s) bound may be required.

The dimerized HC3 helices are located right underneath the dimerization domains, and would not hamper conformational transition between the inward-facing and outward-facing states.

We agree with reviewer that it is important to study functional properties of NHE1 with disrupted HC3 dimer. However, on the one hand, it is challenging to interrupt HC3 dimer mediating by extensive hydrophobic interaction and Van der Waals forces (as shown in Figure 2e). On the other hand, we have to incorporate other biochemical experiments to verify mutations indeed disrupt HC3 dimer. Therefore, we think that exploration of this point is beyond the scope of the present work.

Comment 3) I suspect that this paper was likely written up before the cryo EM structure of NHE9. As such, the paper has more comparisons between NHE1 and the bacterial homologues than it does between NHE1 and NHE9. This is a shame as NHE1 is the first structure of a plasma membrane localised NHE, which is evolutionary quite divergent from NHE9, which is predominantly localised to organelles. An obvious question is what are the structural differences causing the large sequence divergence between these two NHE isoforms? For example, the model of NHE1 based on NHE9 indicated that the intracellular cavity was much more negatively charged than NHE9 (*EMBO J* (2020)39:4541-4559). Furthermore, the NHE9 structure has already established the conserved elevator mechanism with the bacterial Na⁺/H⁺ antiporters and while the NHE1 structures help to refine this model, it nevertheless takes the focus away from the more interesting questions regarding allosteric regulation and inhibition.

Reply: In the NHE9 paper, the authors presented only one inward-facing state and discussed the elevator-like transport mechanism based on comparing with a bacterial homolog. In contrast, we present the first eukaryotic NHE1 in both its inward-facing and inhibitor bound outward-facing

conformations. It is quite natural and important for us to extensively discuss insights on the transport mechanism gained from these NHE1 structures to further refine the elevator mechanism.

Comment 4) For cryo EM processing there are several stages that I found difficult to follow. In ED.Fig.2, they chosen one class from 3D classification and refined, but the CHP1 density has disappeared. Was this density lost due to averaging or masking? Again in ED.Fig.3 the CHP1 density is a bit odd. Before focused classification for CHP1 there is no density for CHP1, but after focused classification the density comes back and looks like an artifact? Please clarify more and make it clear at each step where a mask has been used or not.

Reply: We appreciate the reviewer's comment. The confusion on existence of CHP1 density in the flowchart may come from the cryo-EM maps which were first adjusted to a relative high threshold to clearly show NHE1 transmembrane helices. In this case, the CHP1 density was washed out because of conformational flexibility. In the revised flowchart, we overlaid the same cryo-EM maps with both high and low thresholds together to clearly show simultaneously the CHP1 density and transmembrane helices. Moreover, we have also clarified mask usage in the revised flowchart.

Minor concerns

1. I do not think that the elevator mechanism is “controversial” for electroneutral transporters vs. electrogenic transporters. The elevator mechanism was first proposed by the outward facing state of NapA (Nature (2013)) and confirmed by NapA states in opposite facing conformations (NSMB (2016)). Given the high degree of structural conservation between bacterial electroneutral and electrogenic transporters it is expected all would work with the same type of alternating-access mechanism as recently confirmed (EMBO J (2020)39:4541-4559). Rather than “controversial” I would say the the elevator alternating access mechanism in the NHEs needs to be better refined.

To be clear the only other alternative to an elevator mechanism would be a rocking-bundle model. The difference between a rocking-bundle and elevator mechanism is not decided if the structural transitions “look” more like rocking or vertical movements. The distinction between the two models of alternating access is that in elevator proteins the substrate binds to only ONE of the two domains and then this ONE domain carries the substrate some vertical distance across the membrane, i.e., in contrast to rocking bundles where the substrate binds between two domains and the domains move around the substrate (Annual Review of Biochemistry Vol. 85:543-572). The additional exchanger structures enable one to refine the elevator alternating-access mechanism. The recent cryo EM structure of NHE9 enables some refinement by comparing to bacterial homologues and clearly the NHE1 structures can add more insight, in particular, in regards to allosteric regulation via CHP1 binding to the CTD domain.

Reply: We agree with the reviewer and have removed this statement in our revised manuscript.

Comment 2) Is there any coupling between the inhibitor cariporide and CHP1? Is there still inhibition of NHE1 when the C-terminal tail is truncated? Does the inhibitor conformationally stabilise NHE1? In other words, is part of its ability to inhibit NHE1 caused by the fact that it restricts NHE1 mobility by also interacting with the N-terminus of the neighbouring protomer and also the dimer domain (E346 in TM8)? Does the N-terminal tail change its position when comparing the inhibitor and non-inhibitor bound states?

Reply: The binding sites of the inhibitor cariporide and CHP1 are located on opposite sides of the membrane.

Interactions of different NHE1 antagonists with NHE1 lacking the cytoplasmic C-terminal domain (CTD) have not been examined extensively. However, earlier heterologous expression studies showed that while deletion of the complete CTD greatly reduced NHE1 activity and pH-sensitivity, a fraction of NHE1 was able to traffic to the plasma membrane and retained sensitivity to the NHE inhibitor 5-(N-methyl-N-propyl) amiloride (Wakabayashi, S., et al. (1992). Proc Natl Acad Sci U S A 89(6): 2424-2428). It was concluded the N-terminal transmembrane domain of NHE1 was sufficient both for insertion into the plasma membrane, ion transport and drug sensitivity. Thus, we interpret these findings to indicate that these two sites (drug-binding and CTD) do not have direct coupling.

The critical residues involved in cariporide binding are from both dimerization domain (Y162^{TM3} and E346^{TM8}) and the core domain (E267^{TM6}), and thus the inhibitor binding links the core and dimerization domains and prevents the core domain from translating relative to the dimerization domain. Therefore, we believe that the inhibitor stabilizes the NHE1 dimer at the outward-facing conformational state.

Superimposition of the inhibitor bound and unbound structures demonstrates that there is a little conformational change for the N terminal tail relative to the core domain.

3. A superimposition of the outward and inward facing states results in a 5 Å vertical displacement of the strictly conserved ion-binding D267 based on its Calpha position. This displacement is consistent with the 5Å displacement modelled between inward facing NHE9 and an outward model based on NapA (EMBO J (2020)39:4541-4559). However, NHE9 is less inward facing than inward NapA and, as such, the vertical displacement between opposite facing NapA states is larger at 8Å between Calpha positions and closer to 10Å when the Cbeta positions are included (NSMB (2016)). Part of the reason between larger differences between Calpha and Cbeta positions in NapA is that the ion binding aspartate was found to rotate inwards between outward and inward facing conformations in NapA, which could be mechanistically important, ie., it was proposed that neutralisation of the ion-binding aspartate upon Na⁺ binding would facilitate the transition across the hydrophobic barrier of the dimerization domain surface (NSMB (2016)). Is there any rotation observed of the ion-binding aspartate between opposite facing conformations and what is the Cbeta distance? I understand if this cannot be determined due to a lack of map density for the aspartates (negative scattering factors).

Reply: We appreciate the reviewer's comment. We found that, in addition to the 5 Å translation across the membrane, the D267 also undergoes a 24° rotation. Nevertheless, as the reviewer mentioned, it is not feasible to reliably measure the C_β-C_β distance. We have incorporated the additional rotation of D267 in our revised manuscript (Pg 15 and line 399-401). It reads "Of particular interest, upon the outward- to inward-facing transition, the C_α atom of D267^{TM6} is displaced by 5 Å towards the extracellular side, and its side chain rotates 24°."

4. I was a bit confused by the sentence "However, since the conformational change during the state transition is a relative one between the core and dimerization domains, an alternative movement cannot be ruled out." Is this meant to say that the transport domains stay's fixed and the dimer domains move? Conceptually, this would not make any sense, since the substrate doesnt bind to the dimer domains, but

only the core domains and in this way the substate is carried across the membrane. there is no evidence that ion-binding stabilise the core domains, but the opposite (eLife. 2014; 3: e01412.). Furthermore, from MD simulations of NapA it was modelled the the dimer domain bobs up-and-down a bit, but the transport domains move up and down considerably much more (NSMB (2016). Indeed intrinsic dynamics of NHE9 show that the core domains are very mobile and spontaneous cover 80% of the transitions observed between NapA crystal structures in opposite facing states (EMBO J (2020)39:4541-4559). Also in the evolutionary related citrate transporter structure CitS, there is an asymmetric crystal structures available with one transport domain in the outward conformation and the other in the inward-facing conformation (eLife 2015;4:e09375). In these crystal structures the dimerization domain stays fixed. Taken together, and added to the fact that in the many other elevator proteins the oligomerization interface stays fixed, I think there is no evidence to suggest otherwise.

Reply: We choose to accept the reviewer's comment and have deleted the related content.

5. The salt-bridge between E262 and R425 is conserved in all electroneutral transporters (Nature Communications : 4205 (2018), as modelled and as shown in PaNhaP and NHE9 structures (EMBO J (2020)39:4541-4559). As such, E262 is not thought to be involved directly in ion-binding, but it could still be allosterically coupled to ion-binding. I wouldn't say that the integrity of the E262-R425 salt-bridge is essential for the integrity of the ion-binding domain. In NapA outward and inward facing structures a parallel salt-bridge remains intact between opposite facing states (D156-K305) (NSMB (2016). However, this does not mean that this salt-bridge is "essential" to the core domain integrity as you can make mutations to the lysine and still achieve transport (PNAS February 14, 2017 114 (7) E1101-E1110), its just that most mutations are electroneutral unless the lysine is mutated to histidine. MD simulations in NhaA show that this corresponding salt-bridge is broken upon sodium binding (J Gen Physiol (2014) 144 (6): 529–544.). Its plausible that the E262-R425 salt-bridge is also broken during transport and, at the very least, an intact salt-bridge as seen here does not rule out this possibility.

Reply: We appreciate the reviewer's comment. We have modified our hypothesis on E262-R425 salt-bridge based on reviewer's comment and our structural observation. Now it reads "Moreover, mutation of the conserved E262^{TM6} substantially decreases NHE1 transport activity⁵¹. Whereas it is buried inside the core domain forming a salt bridge with R425^{TM11}, E262^{TM6} is not accessible from either side of the membrane (Figs. S6, S7a, and S7b), suggesting that E262^{TM6} does not directly participate in ion coordination. Nevertheless, further investigation is required for better understanding of its functional role."

6. The proposed rearrangement of TM5b (lines 420-433) doesnt make sense to me and we have not seen this before in any of the previous inward-facing Na⁺/H⁺ antiporter structures. I would argue that the cryo EM density is probably not good enough to assign TM5b geometry properly due to its mobility and its been extended during real-space refinement. A similar issue for this half-helix was also apparent in low resolution NhaA crystal structures (J Gen Physiol (2014) 144 (6): 529–544).

Reply: We feel confident about this structural observation. All of our cryo-EM maps are of high quality with both the main chain and most of side chains clearly resolved (please see Figure S2-S4). Our atomic models are well fitted into the map, evidenced by the model-map correlation coefficients of 0.84, 0.81 and 0.78 for NHE1-CHP1^{Na/6.5}, NHE1-CHP1^{Na/7.5} and NHE1-CHP1^{K/cariporide} complex, respectively. In the methods of structure determination such as crystallography and cryo-EM, an accepted rule of thumb is to think of CC > 0.7 as a good fit and CC < 0.5 as a poor fit (PMID:

30198894). Furthermore, we also have Extended Figures 7e and 7f showing the model-map fitting of TM5b.

7. Was it really the complete full-length NHE1 expression? SDS-page is not sensitive enough to reach this conclusion. Based on the cryo EM structure of NHE9 and sequence analysis it was proposed that NHE1 likely has a cleavable signal peptide rather (EMBO J (2020)39:4541-4559), rather than 14 TMs. Interestingly in the NHE1 structure, residues 1 to 87 aa could not be modelled, which would be the most consistent with this analysis. Do you think NHE1 has a cleaved signal peptide and have you carried our native MS to ascertain otherwise?

Reply: Although we infected HEK293 cell with virus carrying a gene encoding the full-length NHE1, we are uncertain whether the recombinant sample was of full length or some N terminal residues were cleaved as a signal peptide. However, we agree with the analysis in NHE9 paper (EMBO J (2020)39:4541-4559) that suggests that NHE1 has a cleavable signal peptide, which is also consistent with our cryo-EM structures. We also attempted to verify this hypothesis using N terminal sequencing experiment, instead of native MS suggested by the reviewer, because the NHE1 is not uniformly glycosylated. However, Edman degradation was not successful because the N-terminus of NHE1 is blocked.

8. Related to previous question is whether or not the N-terminal tail reaching across to the other promoter (residues 87 to 99) is a form of allosteric coupling between the promoters? As a side note, I would say its unclear if this region and possible regulation is present in NHE9 or not as the first 20 or so amino acids could not be modelled (EMBO J (2020)39:4541-4559). I think the term “structural swapping” is a bit confusing, since its not swapped protein domain, but rather an extended polypeptide that reaches across to the other promoter.

Reply: We accept the reviewer’s comment and have deleted “structural swapping” to avoid confusion.

9. The superimposition is calculated for NHE1 vs NHE9 dimers. What are the RMSD differences between NHE9 vs NHE1 monomers? Does the structural divergence come from differences in oligomerization or local differences within the promoter that could be informative?

Reply: The reviewer raised a good point. The RMSD between NHE9 and NHE1 monomer is ~ 2.1 Å, which is similar to that from the dimer superimposition (~ 2.3 Å), suggesting that the structural divergence is not derived from oligomerization.

We have added this discussion in our revised manuscript (Pg 9 and line 228-232). It now reads “Superimposition of the NHE1 with *Eca*NHE9 and *Pa*NhaP yield RMSDs of ~ 2.2 Å for *Eca*NHE9 dimer (662 C α atom-pairs), ~ 2.1 Å for *Eca*NHE9 protomer (324 C α atom-pairs), ~ 2.6 Å for *Pa*NhaP dimer (664 C α atom-pairs) and ~ 2.3 Å for *Pa*NhaP protomer (337 C α atom-pairs), respectively, suggesting that the structural difference is mainly derived from the NHE protomer.”

10. In NHE9 it was proposed that PIP2 lipids bind at the hydrophobic interface to stabilise the homodimer. However, the PIP2 binding seemed to require an loop domain, which is not present in NHE1. Is there any clues as to what lipids bind at the interface?

Reply: In our cryo-EM maps, we did identify six strip-shaped density in the same extracellular cavity (Figures 2c, S4e, S5f and S6e), and thus suspect that it represents hydrophobic tails of some lipid molecules, which appear to play a stabilization role in the NHE1 dimer. Nevertheless, because of lack of density for their head groups, we are unable to determine the identity of these lipid molecules bound in the cavity.

In regard to the paper by Winkelmann et al. (2020), they identified a role of negatively-charged lipids, such as phosphoinositides PIP₂ and PIP₃, in stabilizing the dimeric structure of horse NHE9 in reconstituted liposomes (as assessed by a thermo-stabilization assay and native mass spectrometry). Notably, three lysine residues in the TM2-TM3 loop domain positioned above the dimer interface domain and oriented towards the endosomal lumen were found to be critical for PIP₂-mediated stabilization. It was concluded that phosphoinositides bind to the TM2-TM3 loop domain to stabilize the functional dimer. While this *in vitro* observation is intriguing, it is difficult to reconcile with the native localization of phosphoinositides in intact cells. Phosphoinositides such as PIP₂ and PIP₃ are normally located in the membrane leaflet facing the cytoplasm, not the membrane leaflet facing the endosomal lumen or extracellular space (Di Paolo, G. and De Camilli, P., 2006. *Nature*, 443(7112), pp.651-657; Balla T. *J Physiol.* 2007;582(Pt 3):927-937). Hence, the role of phosphoinositides in NHE9 structure and function are still unclear and will require further investigation to resolve.

11. Related to the above question PIP₂ is reported to be bind at IL7. I found it difficult to follow the authors arguments of how PIP₂ binding could increase NHE1 activity.

Reply: Currently, we simply do not know how PIP₂ binding increases NHE1 activity. The PIP₂ bound complex structures are likely to be required to fully understand the modulation roles of PIP₂. We have clarified this point in the revised manuscript (Pg 17, line 470-476). It reads "...So far, residues on IL7, IL8, and HC2 have been identified to be critical for PIP₂ modulation^{54,55}, which are all close to the CHP1 binding site. Hence, we proposed that PIP₂ modulates the activity of NHE1 by affecting CHP1 conformation, particularly for the CHP1 in the inward-facing conformation which exhibits highly motility relative to NHE1 in the absence of PIP₂. The structure of a PIP₂-bound NHE1-CHP1 complex is required to fully understand the modulation mechanism of PIP₂ on NHE1 transport activity.".

12. Please include the following reference first proposing an elevator mechanism for Na⁺/H⁺ exchangers and indeed the first evidence that an elevator model is a general type of alternating-access mechanism for secondary-active transport; as first observed in the glutamate transporter GltPh. "A two-domain elevator mechanism for sodium/proton antiport". *Nature*. 2013 Sep 26; 501(7468): 573–577.

Reply: We agree and have updated the reference.

13. I noticed that the first low resolution cryo EM electron crystallography structure of NhaA was referenced instead of the NhaA crystal structure. I think the later is a more appropriate reference as it enabled the first glimpse into the overall NhaA-fold as shared by all Na⁺/H⁺ exchangers (*Nature* 435 (2005), 1197–1202) and even Na⁺-coupled bile acid symporters (*Nature*. 2011 Oct 5; 478(7369): 408–411).

Reply: We agree and have added the reference of the crystal structure of NhaA.

REVIEWERS' COMMENTS

Reviewer #1 (Remarks to the Author):

The authors have thoroughly addressed the great majority of the concerns raised by the reviewers, and I find the manuscript much improved. I am still of the opinion that demonstrating function in an in vitro transport assay is needed to unequivocally show that the protein is in its native conformation. But I respect the author's explanation that their low yield of the complex renders this unfeasible, and I support the publication of this important work.

Reviewer #2 (Remarks to the Author):

In my opinion the authors managed to answer the critical proposals and questions of my co-reviewers satisfactorily.

However, line 165 should read "cysteine accessibility" and not "cystine accessibility"

Reviewer #3 (Remarks to the Author):

I thank the reviewers for their revised manuscript. However, we seem to be mis-communicating somewhat. The proposed mechanism for CHP1 activation is still unclear to me. I might be just me, but I expect other readers might have similar difficulties with teasing out a mechanism from the current paper.

My question is as follows: Based on the current NHE1-CHP1 structures how do you envision CHP1 allosterically increasing NHE1 ion-exchange activity?

You have written that you think HC3-HC3 interactions are independent of the conformational state. This makes sense to me. But, then you propose that CHP1 forms a more stable complex in the outward-facing conformation and that the CHP1-HC3 interaction is important for conformational transition from the inward-facing state to the outward-state. Is this not a contradiction? Could it not be because the stability of CHP1 is linked to HC3-HC3 interactions, since they are both on the CTD domain? In other words, is it not equally plausible that CHP1 is more stable in the outward-facing conformation because the CTD is stabilised by HC3-HC3 interactions?

One could argue that HC3-HC3 interactions have formed because NHE1 is less dynamic overall in the presence of the bound inhibitor. To be clear, I understand that the location of the inhibitor binding site is a long way from the location of the CHP1 binding site, but the protein is very dynamic and just by stabilizing the transporter the CTD could now form interactions that are less spontaneously populated w/o an inhibitor present. Another way to rephrase this question is, if one was now able to obtain the structure of NHE1 conformationally stabilised in the inward-facing state by an inward-facing specific inhibitor, do you think the CTD would look the same as the current structure ... or could the CTD rearrange as in the outward-facing state observed here?

On the other hand, if you are of the opinion that the differences in the structure of the CTD domain-CHP1 complexes are an "exact" physiological reflection of outward-facing vs. inward-facing states formed during alternating access, what is the proposed mechanism for CHP1 activation? Furthermore, Na⁺/H⁺ exchangers are one of the fastest known secondary-active transporters with rates around 1000 ion/sec. Would it not be energetically unfavourable for these large rearrangements to take place in the CTD between outward and inward-facing states? Can you really conclude that CHP1 molecules differentially associate with the two major states of the NHE1 dimer without a structure of either i) NHE1 in the outward-facing state w/o inhibitor bound or ii) NHE1 in the inward-facing state with an

inhibitor bound?

From my point of view, I think the paper would be improved with a just few summary sentences of how the authors think that CHP1 and the CTD is coupled to the elevator structural transitions.

Congratulations again - the structure of NHE1 in complex with CHP1 is a fanatic accomplishment!

Minor points:

1. Referee 1 asked that the authors demonstrate that purified human NHE1 is functional by demonstrating activity in proteolipsomes. Whilst I agree this is typically the requirement for structural studies of transporters, I think the fact the inhibitor binds is demonstration that human NHE1 is well-enough folded; at least in our hands, its taken more than several years to get ion-exchange activity of NHEs in proteolipsomes and, as such, I do not think its not something that can be done in a short time-frame.

2. In regards to the topology, I think it would be helpful to point out that the length of the N-terminus is correlated with the presence of the predicted signal peptide in the NHEs (EMBO J (2020)39:4541-4559). NHE9, for example, does not have a predicted signal peptide as it naturally has a shorter N-terminal tail. I think the conserved structures between NHE9 and NHE1 is also strong evidence that the first region is a cleavable signal-peptide in NHE1.

The reason for the requirement of the signal peptide is that N-terminal translocation becomes inefficient as the N-terminal tail gets too long and/or charged. This is why GPCRs with long N-terminal tails have a signal peptide so that the N-terminal instead fold into the ER lumen. Notably, the fact that NHE1 and other isoforms requires this charged and glycoylated domain does indicate that its likely to have some functional role.

3. I still think it is confusing that mammalian NHE9 from *Equus caballus* is referred to as "EcaNHE9". I think it should just be written as "NHE9" or "horse NHE9" so its not confused with a bacterial protein. At least, I am not aware that the term "NHE9" can only be used in reference to the human homologue.

Reviewer #3 (Remarks to the Author):

I thank the reviewers for their revised manuscript. However, we seem to be mis-communicating somewhat. The proposed mechanism for CHP1 activation is still unclear to me. I might be just me, but I expect other readers might have similar difficulties with teasing out a mechanism from the current paper.

My questions is as follows: Based on the current NHE1-CHP1 structures how do you envision CHP1 allosterically increases NHE1 ion-exchange activity?

You have written that you think HC3-HC3 interactions are independent of the conformational state. This makes sense to me. But, then you propose that CHP1 forms a more stable complex in the outward-facing conformation and that the CHP1-HC3 interaction is important for conformational transition from the inward-facing state to the outward-state. Is this not a contradiction? Could it not be because the stability of CHP1 is linked to HC3-HC3 interactions, since they are both on the CTD domain? In other words, is it not equally plausible that CHP1 is more stable in the outward-facing conformation because the CTD is stabilised by HC3-HC3 interactions?

One could argue that HC3-HC3 interactions have formed because NHE1 is less dynamic overall in the presence of the bound inhibitor. To be clear, I understand that the location of the inhibitor binding site is a long way from the location of CHP1 binding site, but the protein is very dynamic and just by stabilizing the transporter the CTD could now form interactions that are less spontaneously populated w/o an inhibitor present. Another way to rephrase this question is, if one was now able to obtain the structure of NHE1 conformationally stabilised in the inward-facing state by an inward-facing specific inhibitor, do you think the the CTD would look the same as the current structure. or could the CTD rearrange as in the outward-facing state observed here?

On the other hand, if you are of the opinion that the differences in the structure of the CTD domain-CHP1 complexes are a “exact” physiological reflection of outward-facing vs. Inward-facing states formed during alternating access, what is the proposed mechanism for CHP1 activation? Furthermore, Na⁺/H⁺ exchangers are one of the fastest known secondary-active transporters with rates around 1000 ion/sec. Would it not be energetically unfavourable for these large rearrangements to take place in the CTD between outward and inward-facing states? Can you really conclude that CHP1 molecules differentially associate with the two major states of the NHE1 dimer without a structure of either i) NHE1 in the outward-facing state w/o inhibitor bound or ii) NHE1 in the inward-facing state with an inhibitor bound?

From my point of view, I think the paper would be improved with a just few summary sentences of how the authors think that CHP1 and the CTD is coupled to the elevator structural transitions.

Congratulations again - the structure of NHE1 in complex with CHP1 is a fanatic accomplishment!

Reply: We appreciate the reviewer for sharing his/her thoughts. Indeed, based on the current structures, we cannot firmly establish the mechanism by which CHP1 enhances NHE1 activity. However, our structures agree with many previous biochemical data, and we believe that the new

structural information will stimulate further studies to solve the mysteries on the allosteric regulation of NHE1 by CHP1.

We agree that that the extent of the rearrangements may not precisely reflect the situation in situ. It is conceivable that the actual magnitude of this conformational change may be much smaller and subtler in situ. And we speculate that the conformational heterogeneity of NHE1-CHP1 complexes shown in the inward-facing conformational state would probably be reduced by binding of PIP2 or other regulatory partners. To fully understand the CHP1 modulation mechanism, we probably need to determine more complete complex structures, for instance in the presence of the PIP2 or other associated proteins. We briefly discuss this speculation in the manuscript. It reads “So far, residues on IL7, IL8, and HC2 have been identified to be critical for PIP2 modulation^{54,55}, which are all close to the CHP1 binding site. Hence, we propose that PIP2 modulates the activity of NHE1 by affecting CHP1 conformation, particularly for CHP1 in the inward-facing conformation which exhibits high motility relatively to NHE1 in the absence of PIP2. The structure of a PIP2-bound NHE1-CHP1 complex is required to fully unveil the modulation mechanism of PIP2 on NHE1 transport activity.”

We prefer to think that the stability of CHP1 is not required for the dimerization of the HC3 helices. This is because the HC3-HC3 interaction is constitutively present in isolated CTD (PMID: 15323573); in contrast, the CHP1 subunits are stabilized only in the outward-facing conformational state. Therefore, it is likely that the conformational stability of CHP1 in the inward-facing state is more or less correlated with the conformation of the NHE1 dimer. According to the current elevator model, in the outward state, the core domain moves towards extracellular side across the membrane. It results in a new intracellular surface that is able to accommodate CHP1 proximal to NHE1. The interaction of CHP1-HC3 will further stabilize this conformation.

Regarding the relationship between inhibitor binding and CTD conformation of NHE1, we believe that the CTD conformation change is affected by rearrangements of the transmembrane domain of NHE1 during ion exchange, instead of by binding of a specific inhibitor. That being said, we agree with the reviewer that it would be instructive to obtain a structure of NHE1-CHP1 in the outward-facing state without the inhibitor bound. Perhaps this could be achieved by preparing samples in the presence of PIP2 and across a broader range of pH conditions. These possibilities will form the basis for future studies. If successful, this information could help further refine our mechanistic understanding of how CHP1 allosterically regulates NHE1.

Now we have added a more specific discussion to clarify our thoughts about modulation mechanism of CHP1 on page 18, line 492. Now it reads “we speculate that this CHP1–HC3 interaction is important for conformational transition from the inward-facing state to the outward-facing state. However, to fully interpret how CHP1 enhance the NHE1 activity, a more complete complex, in the presence of PIP2 or other regulatory proteins, seems to be required.”

Minor points:

1. Referee 1 asked that the authors demonstrate that purified human NHE1 is functional by demonstrating activity in proteoliposomes. Whilst I agree this is typically the requirement for structural studies of transporters, I think the fact the inhibitor binds is demonstration that human NHE1 is well-enough folded; at least in our hands, its taken more than several years to get ion-exchange activity of NHEs in proteoliposomes and, as such, I do not think its not something that can be done in a short time-frame.

Reply: We thank reviewer for providing a supporting opinion.

2. In regards to the topology, I think it would be helpful to point out that the length of the N-terminus is correlated with the presence of the predicted signal peptide in the NHEs (EMBO J (2020)39:4541-4559). NHE9, for example, does not have a predicted signal peptide as it naturally has a shorter N-terminal tail. I think the conserved structures between NHE9 and NHE1 is also strong evidence that the first region is a cleavable signal-peptide in NHE1.

The reason for the requirement of the signal peptide is that N-terminal translocation becomes inefficient as the N-terminal tail gets too long and/or charged. This is why GPCRs with long N-terminal tails have a signal peptide so that the N-terminal instead fold into the ER lumen. Notably, the fact that NHE1 and other isoforms requires this charged and glycoylated domain does indicate that its likely to have some functional role.

Reply: We appreciate the reviewer's comment. We have incorporated this view and related reference to support the NHE1 might harbors a cleavable signal peptide. It reads "...which is predicted to be a cleavable signal peptide using the SignalP-5.0 server³⁶, in line with a concept that a large extracellular N-terminal generally harbors a cleavable signal peptide^{19,37}".

3. I still think it is confusing that mammalian NHE9 from *Equus caballus* is referred to as "EcaNHE9". I think it should just be written as "NHE9" or "horse NHE9" so its not confused with a bacterial protein. At least, I am not aware that the term "NHE9" can only be used in reference to the human homologue.

Reply: We have changed EcaNHE9 to horse NHE9 throughout the manuscript.